# Cross-protection and cross-feeding between *Klebsiella pneumoniae* and *Acinetobacter baumannii* promotes their co-existence

Lucie Semenec [1,2,5], Amy K. Cain [1,2,5], Catherine J. Dawson [3,5], Qi Liu[2], Hue Dinh [1,2], Hannah Lott[2], Anahit Penesyan [1,2], Ram Maharjan[1,2], Francesca L. Short [4], Karl A. Hassan [1,3,6] ✉ & Ian T. Paulsen [1,2,6] ✉

*Acinetobacter baumannii* and *Klebsiella pneumoniae* are opportunistic pathogens frequently co-isolated from polymicrobial infections. The infections where these pathogens co-exist can be more severe and recalcitrant to therapy than infections caused by either species alone, however there is a lack of knowledge on their potential synergistic interactions. In this study we characterise the genomes of *A. baumannii* and *K. pneumoniae* strains co-isolated from a single human lung infection. We examine various aspects of their interactions through transcriptomic, phenomic and phenotypic assays that form a basis for understanding their effects on antimicrobial resistance and virulence during co-infection. Using co-culturing and analyses of secreted metabolites, we discover the ability of *K. pneumoniae* to cross-feed *A. baumannii* by-products of sugar fermentation. Minimum inhibitory concentration testing of mono- and co-cultures reveals the ability for *A. baumannii* to cross-protect *K. pneumoniae* against the cephalosporin, cefotaxime. Our study demonstrates distinct syntrophic interactions occur between *A. baumannii* and *K. pneumoniae*, helping to elucidate the basis for their co-existence in polymicrobial infections.

Polymicrobial infections caused by two or more pathogenic microorganisms, although relatively common, are largely understudied. Clinical diagnosis of bacterial infections[1] often only consider the predominant infecting microorganism and may overlook pathogens present in lower abundance. Our knowledge of pathogens largely comes from pure culture laboratory studies, which have been pivotal to understanding single species infections, but have given little information on coinfection dynamics. Given that minority populations of bacterial communities can have significant impacts on the physiology and behaviour of dominant members[2,3], it is important to understand possible interspecies interactions between coinfecting pathogens and

the effects they may have on virulence and antibiotic resistance within polymicrobial infections.

Various studies have found that polymicrobial infections can lead to increased virulence and antimicrobial resistance[4,5]. Dual-infections by *Acinetobacter baumannii* or *Pseudomonas aeruginosa* in patients with carbapenem-resistant *Enterobacteriaceae* (CRE) have been shown to have increased antibiotic resistance levels and mortality rates compared to single infections[6]. For instance, increased mortality has been observed in critically ill patients co-infected with *Klebsiella pneumoniae*, *P. aeruginosa* and/or *A. baumannii*[6]. Furthermore, co-infection with multi-drug resistant (MDR) *Acinetobacter* spp. and

[1]ARC Centre of Excellence in Synthetic Biology, School of Natural Sciences, Macquarie University, North Ryde, NSW 2113, Australia. [2]School of Natural Sciences, Macquarie University, North Ryde, NSW 2113, Australia. [3]School of Environmental and Life Sciences, University of Newcastle, Callaghan, NSW 2308, Australia. [4]Department of Microbiology, Biomedicine Discovery Institute, Monash University, Clayton, VIC 3800, Australia. [5]These authors contributed equally: Lucie Semenec, Amy K. Cain, Catherine J. Dawson. [6]These authors jointly supervised this work: Karl A. Hassan, Ian T. Paulsen. ✉e-mail: karl.hassan@newcastle.edu.au; ian.paulsen@mq.edu.au

extended spectrum β-lactamase (ESBL) producing microorganisms (*K. pneumoniae* and *Escherichia coli*) has been identified in ca. 38% of hospitalised patients infected with MDR *Acinetobacter* spp[6–8].

In this study, we investigated interactions between two bacterial pathogens, *A. baumannii* strain AB6870155, and *K. pneumoniae* strain KP6870155 that were co-isolated from a single lung infection. *A. baumannii* and *K. pneumoniae* are opportunistic human pathogens that are both involved in a range of similar infections, including respiratory, urinary tract and blood infections, particularly in immuno-compromised patients. Both have been classified as members of the ESKAPE pathogens, the six top priority dangerous multidrug resistant microorganisms (*Enterococcus faecium*, *Staphylococcus aureus*, *Klebsiella pneumoniae*, *Acinetobacter baumannii*, *Pseudomonas aeruginosa*, and *Enterobacter* species) by the Infectious Diseases Society of America and World Health Organization[9,10].

The virulence mechanisms of *A. baumannii* and *K. pneumoniae* have been extensively studied over the years in the context of each individual species. Both pathogens often form biofilms in patient lungs[11,12] which protects against nutrient limitation, predation and osmotic stress, antibiotic treatment and consequently gives them remarkable resilience[13,14]. Both strains also produce siderophores that enable efficient iron uptake in host environments limited in iron[15,16], and iron-dependent super oxide dismutase enzymes (SodB) that can neutralise reactive oxygen species (ROS) generated by the Fenton reaction[17]. There are also some differences in the virulence strategies of these two pathogens. *A. baumannii* is capable of polyamine synthesis, which plays a role in virulence and biofilm formation in respiratory tract pathogens[18]. 1,3-diaminopropane (DAP) is the predominant polyamine produced by *A. baumannii* which coincidentally binds to *Acinetobacter* siderophores[19]. Polyamine synthesis in *K. pneumoniae* is slightly more diverse, involving production of putrescine and cadaverine[20] and it is armed with transporters that can efflux these cationic hydrocarbons out of the cell[11]. In terms of their interactions with host cells, *K. pneumoniae* produces two well characterised cell surface polysaccharides, lipopolysaccharide O antigen and polysaccharide capsule (K) and less well elucidated enterobacterial common antigen, which allow it to evade host immune attacks[21,22]. *A. baumannii* forms lipooligosaccharide and capsule but lacks lipopolysaccharide, due to the absence of an O-antigen ligase[23]. The *A. baumannii* capsular polysaccharide (encoded by the K locus) promotes biofilm formation and enables it to withstand desiccation[24]. *A. baumannii* counteracts host immune responses differently from *K. pneumoniae*, through expression of OmpA and Omp33 proteins which allow it to survive within host autophagosomes by preventing complete autophagy[25] or concentrating OmpA on outer membrane vesicles[26]. Overall, these two pathogens have a variety of overlapping and distinct virulence mechanisms within the host.

Despite the co-existence of *A. baumannii* and *K. pneumoniae* in a variety of life-threatening infections and their diverse array of virulence mechanisms, no study to date has focused on their interaction dynamics. In this study we sought to investigate these two species isolated from a single lung infection in order to understand their potential interaction dynamics within the host environment which may potentiate virulence and antibiotic resistance. Their genomes were sequenced and assembled using short and long-read sequencing and genomic analyses were performed to characterise their evolution, plasmid profiles, serotypes, resistomes, and repertoire of transporters. We performed transcriptomic analysis on the two co-isolates, AB6870155 and KP6870155, grown in dual-species co-culture biofilms and compared that to their gene expression in their respective mono-culture grown biofilms in synthetic lung mimicking medium (SLMM). We performed phenotypic assays to study what effects their interaction had on their physiology, metabolism, biofilm formation and antimicrobial responses in vitro using SLMM. In addition, we studied their virulence in vivo in the context of co-infections to lay the groundwork for developing treatment strategies for these two important ESKAPE pathogens and polymicrobial infections more generally.

## Results

### Genomic and phylogenetic characterisation of AB6870155 and KP6870155

The genomes of *A. baumannii* AB6870155 and *K. pneumoniae* KP6870155 were sequenced using a combination of Oxford Nanopore long-read and Illumina short-read sequencing. The genome of AB6870155 comprises of a 4.0 Mb chromosome and three plasmids, pAB0155_1-3 (Fig. S1), with sizes ranging from 1.8–8.7 kb (Table 1). The KP6870155 genome consists of a 5.36 Mb chromosome and five plasmids termed pKP0155_1-5 (Fig. S2), ranging in size from 2.1–67.3 Kb (Table 1). The KP6870155 plasmids were of IncFIB(K) (pKP0155_1), IncU (pKP0155_2), IncR (pKP0155_3), IncN (pKP0155_4) and Col440I (pKP0155_5) incompatibility types as determined by PlasmidFinder[27] and contained various resistance genes and plasmid maintenance and mobilisation genes (Figs. S2–S3, Supplementary Data 1–2). None of the cryptic AB6870155 plasmids belonged to any of the known incompatibility types.

Phylogenetic analyses placed AB6870155 within the International Clone I (IC1) clonal complex strains, being most closely related to the A85 strain, also a human sputum isolate[28] (Fig. 1a). Like A85, AB6870155 also contains a Tn*6168* IS*Aba1*-bounded transposon carrying an *ampC* gene[28,29]. AB6870155 also harbors *gyrA* and *parC* mutations conferring fluoroquinolone resistance as found in A85 but only contains one of the two carbapenem resistance genes, *oxa-51* not *oxa23* (Supplementary Data 3). The KP6870155 genome was most closely related to a virulent strain, Bckp186, of *K. pneumoniae* that was isolated in China from a dairy cow with acute mastitis (Fig. 1b). Kleborate[30] identified KP6870155 as a sequence type 8 (ST8) strain, which is not a well-known MDR lineage, possessing a type KL3 capsule locus and O-antigen locus O1v2.

To understand the evolution of these strains, we used PPanG-GOLiN software[31] to compute the pangenome of *A. baumannii* AB6870155 and 172 complete *A. baumannii* genome sequences available in RefSeq, downloaded on September 30, 2020 (Supplementary Data 4). Of the total number of genes in AB6870155, 12.4% were cloud (accessory/dispensable) genes while the average number of cloud genes in the *A. baumannii* genome was only 8.1% (Fig. S3). A *K. pneumoniae* pangenome was also constructed using KP6870155 and 530 complete *K. pneumoniae* genome sequences available in RefSeq downloaded on September 30, 2020 (Supplementary Data S5). There were also more cloud genes, 6.5%, in the KP6870155 strain as compared to the average in the *K. pneumoniae* genome, 4.2% (Fig. S3).

Given that both strains had a relatively large repertoire of accessory genes, we determined which genes were located within regions of genome plasticity (RGPs) using PPanGGOLiN, which defines a RGP as a set of consecutive genes that are part of the shell (core) or cloud genomes. *A. baumannii* AB6870155 contained 33 RGPs, one of which was located on pAB0155_1, and *K. pneumoniae* KP6870155 contained 39 RGPs, 35 on the chromosome and an RGP on each of the plasmids (pKP0155_1-4 but not in pKP0155_5) (Supplementary Data 1 and 6). Of the genes located in the AB6870155 RGPs, 59% were hypothetical genes and 62% were hypothetical genes for KP6870155. The roles of these

## Table 1 | Genomic data and assembly statistics for *A. baumannii* AB6870155 and *K. pneumoniae* KP6870155

| Strain | Genome size (bp) | GC (%) | *No. plasmids: sizes (Kb) | No. genes | No. tRNA |
|---|---|---|---|---|---|
| AB6870155 | 3,990,368 | 39.24 | **3**: 8.7, 2.3, 1.8 | 3902 | 75 |
| KP6870155 | 5,364,104 | 57.35 | **5**: 67.3, 62.6, 51.5, 6.6, 2.1 | 5476 | 89 |

*Number of plasmids are denoted in bold text.

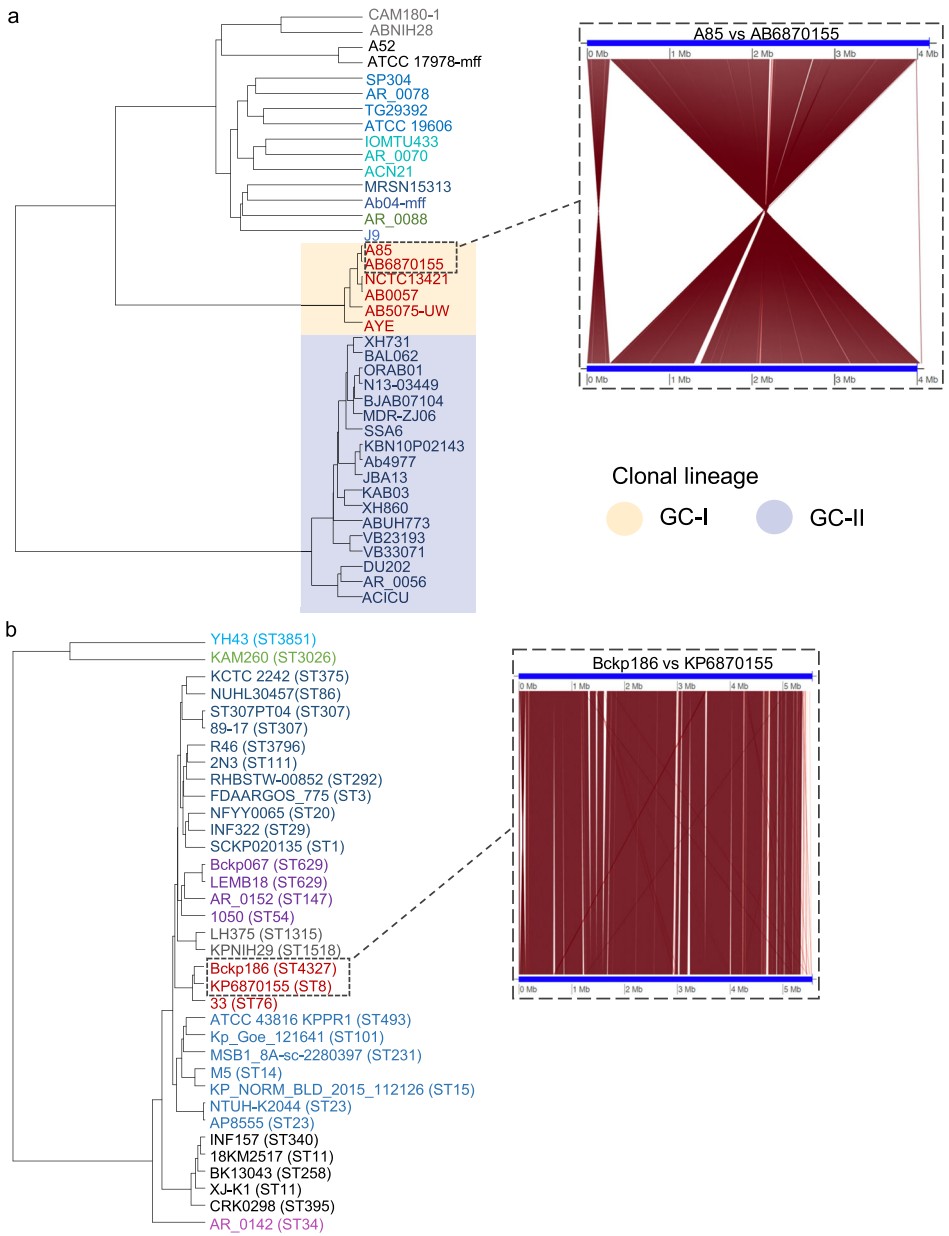

**Fig. 1 | Phylogeny and synteny of *A. baumannii* AB6870155 and *K. pneumoniae* KP6870155.** Dendrograms representing Euclidean distance matrices from pairwise fastANI distances between genomes of **a** AB6870155 and other *A. baumannii* strains and, **b** KP6870155 and other *K. pneumoniae* genomes. Panel inserts show mappings of orthologous regions between each strain and its closest relative strain computed by fastANI and plotted with genoPlotR[125]. *K. pneumoniae* sequence types are in parentheses next to the strain name. Source data are provided as a Source Data file.

genes may have not yet been identified in coinfection adaptation which is a limitation of this work and warrants further study.

## Co-culturing KP6870155 and AB6870155 benefits growth of *A. baumannii* in synthetic lung mimicking medium (SLMM)

To uncover the molecular mechanisms behind co-culture interactions we performed RNA-seq on mono- and mixed-species biofilm cultures grown in SLMM, which mimics the nutritional environment of the respiratory tract from which they were isolated. In the mixed-species co-cultures, 19,846,556 RNA-seq reads were mapped to either AB6870155 or KP6870155; of these ca. 82% mapped to AB6870155 and the remaining to KP6870155, which indicates a higher proportion of *A. baumannii* in the co-culture biofilms at the 24-h timepoint when RNA was harvested. After normalising RNA read counts, AB6870155 grown

in co-culture versus pure culture had 368 genes significantly (adj. $p \leq 0.05$ and $|log2FC| \geq 1$) higher expression levels and 353 genes with significantly lower expression levels (Supplementary Data 2). Genes involved in energy metabolism, including glycolysis, TCA cycle, pentose phosphate pathway, respiration and ATP biosynthesis had overall increased expression in co-culture grown AB6870155 relative to mono-cultured AB6870155 (Supplementary Data 7, Fig. 2a). Additionally, pathways involved in cellular processes (cell cycle and division, genetic transfer, biofilm, adhesion, locomotion and response to virus), stimulus response (starvation, temperature, DNA damage, osmotic stress and oxidant detoxification), central dogma (transcription, translation, DNA/RNA/protein metabolism and protein folding/secretion) and other pathways (Supplementary Data 7) had overall increased expression levels in AB6870155 when co-grown with *K. pneumoniae*

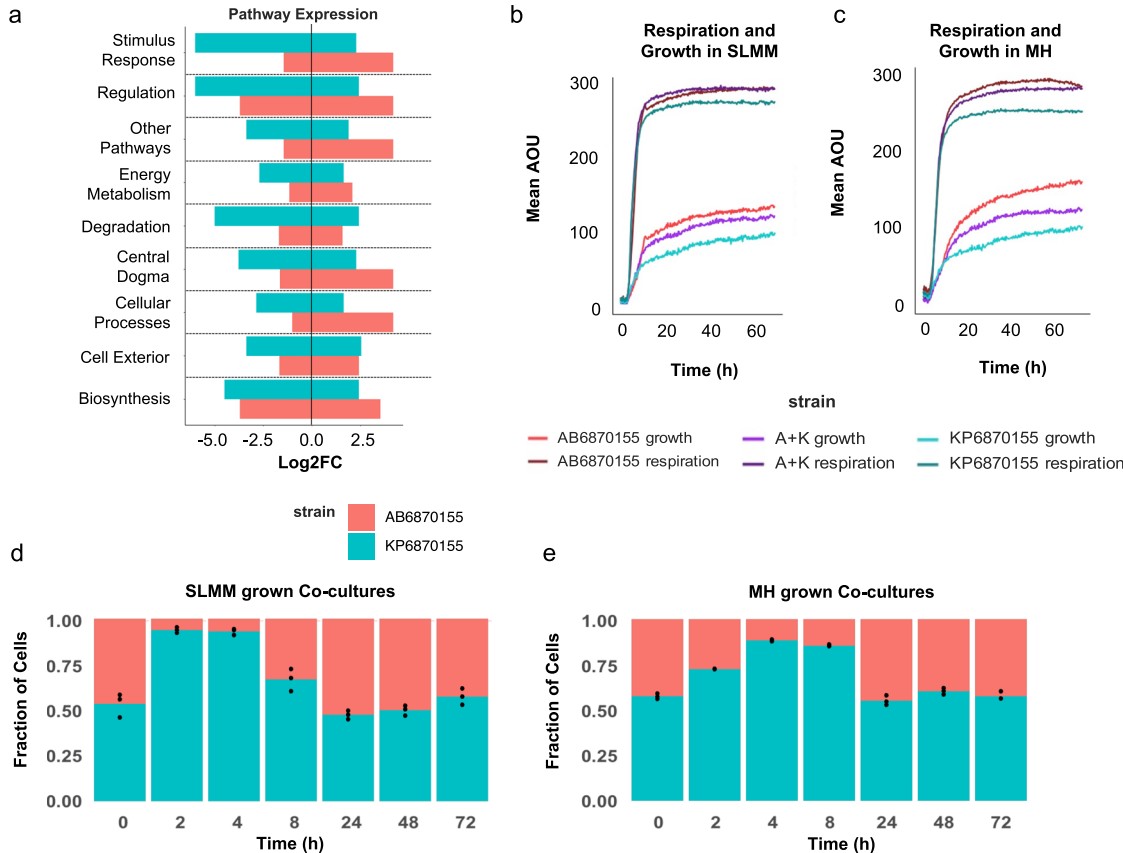

**Fig. 2 | Respiration, growth and gene expression of AB6870155 and KP6870155.**
**a** Distribution of gene expression across major pathways as measured by the log 2-fold change (Log2FC) of gene expression in co-cultures of each strain versus their pure culture counterparts grown in SLMM and harvested for RNA at the 24 h timepoint. **b** Co-culture AB6870155 + KP6870155 (A + K) and mono-culture growth and respiration of *A. baumannii* AB6870155 and *K. pneumoniae* KP6870155 grown in SLMM and **c** MH media as measured in arbitrary omnilog units (AOU). **d** Co-culture proportions of AB6870155 and KP6870155 during growth in SLMM (*n* = 3 independent bacterial cultures), and **e** MH media (*n* = 3) as measured by qPCR. Source data are provided as a Source Data file.

versus mono-cultured, indicating a likely improvement to its growth rate during co-culturing (Fig. 2a). Conversely, in co-cultures, the transcripts of 431 KP6870155 genes were significantly more abundant including those involved in cell exterior and biosynthesis, and 529 gene transcripts were significantly less abundant, including genes involved in stimulus response and regulation (Supplementary Data 8).

To further investigate the apparent fitness advantage of *A. baumannii* AB6870155 grown in co-culture with *K. pneumoniae* KP6870155, we monitored their respective growth rates, abundance and metabolic activity within planktonic mono- and co-cultures in Mueller–Hinton (MH) media and SLMM. *A. baumannii* AB6870155 mono-cultures demonstrated a significant fitness advantage (area under curve (AUC)-AUC 7612) over *K. pneumoniae* KP6870155 mono-cultures (AUC 4877) and co-cultures (AUC 6710) when grown in MH media (Fig. 2c). To determine the proportions of each strain within the co-cultures, we performed species-specific qPCR at various timepoints during growth. Co-cultures grown in MH media strongly favoured growth of *K. pneumoniae* during lag to exponential phase, potentially due to its enhanced utilisation for rich carbon sources over *A. baumannii* (Fig. 2e). There was a shift toward increased *A. baumannii* towards stationary phases of growth (>20 h), possibly due to consumption of *K. pneumoniae* metabolic by-products. Interestingly, *A. baumannii* fared better in SLMM grown co-cultures, where the proportion of AB6870155 began to increase during exponential phase (Fig. 2d), while this shift only happened at stationary phase (24 h) when grown in MH. *A. baumannii* AB6870155 mono-cultures had a less pronounced growth advantage (AUC 6819) over *K. pneumoniae* KP6870155 mono-cultures (AUC 4856)

when grown in SLMM (Fig. 2b) and the respiratory activity of co-cultures (AUC 17439) was higher than that of AB6870155 mono-cultures in SLMM (AUC 17033) (Fig. 2b) which corresponded with the increased expression of respiratory pathways in *A. baumannii* co-cultured with *K. pneumoniae*. It is likely that most of the respiratory activity in co-cultures can be attributed to *A. baumannii* given its higher expression of aerobic respiration pathways in the SLMM co-culture grown biofilms (Fig. S4).

This suggests a possible change in nutrients available throughout co-culture growth that allows the co-culture populations to fluctuate in favour of AB6870155 over time in nutritionally stringent versus nutrient-rich conditions.

### Carbon source profiling of AB6870155 and KP6870155 demonstrates higher metabolic flexibility of *K. pneumoniae*

To elucidate why certain media favoured the growth and metabolic activity of each organism in co-cultures, phenotype microarrays (PMs) were performed with Biolog PM plates (PM01-02). This facilitated high throughput probing into the utilisation of 190 different carbon sources by the two strains. *K. pneumoniae* KP6870155 demonstrated significantly higher metabolic flexibility than *A. baumannii* AB6870155 (Fig. 3a), being able to utilise 101 of the different carbon sources tested, whereas AB6870155 could only utilise 50 (Fig. 3b, Supplementary Data 9). Metabolic versatility of KP6870155 is in line with its larger genome that accommodates a greater repertoire of putative transport proteins (886) compared to AB6870155 (402), and genes encoding metabolic pathways not found in AB6870155, like the *dha* genes

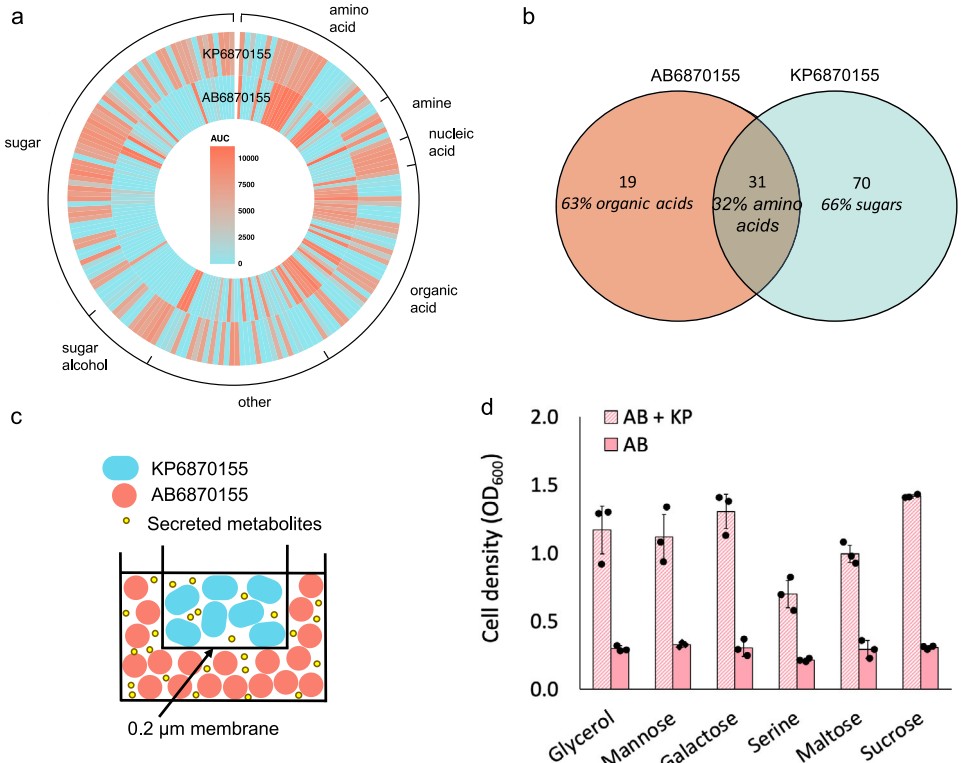

**Fig. 3 | Carbon source utilisation and cross-feeding between *A. baumannii* AB6870155 and *K. pneumoniae* KP6870155. a** Carbon-source utilisation activity calculated by AUC of AB6870155 (inner ring) and KP6870155 (outer ring) based on PM01-02 Biolog Phenotype Microarrays. **b** Venn diagram of C-source utilisation from phenome data showing overlap between AB6870155 and KP6870155. **c** Schematic of cross-feeding experiment utilising Millicell culture inserts (Merck). **d** Growth of AB6870155 in Millicell plates with minimal media and select compounds (glycerol, mannose, galactose, serine, maltose, sucrose) provided as the sole carbon source in the presence (denoted as AB + KP) and absence of KP6870155 (denoted as AB) (*n* = 3 independent bacterial cultures). Data are presented as mean values + /− SD. Source data are provided as a Source Data file.

required for glycerol degradation and genes encoding beta-galactosidases involved in the hydrolysis of galactosides. Of the carbon sources accessible to *A. baumannii*, 63% were organic acids, while *K. pneumoniae* KP6870155 predominantly utilised sugars (Supplementary Data 9; Fig. S5). The two strains shared a capacity for utilising 31 carbon sources, many of which were amino acids and organic acids (Fig. 3b). Although both were able to respire using amino acids as a carbon/energy source, *A. baumannii* had a higher respiratory activity when utilising these compounds (Fig. 3a) while *K. pneumoniae* had highest respiratory activity when grown on readily available carbon sources like sugars. Given the different metabolic fingerprints of these two strains, it is possible that they may physiologically complement one another. Alternatively, given their metabolic differences, the strains may inhabit different microenvironments in the human respiratory tract during infection.

Inspection of the RNA-seq data showed that KP6870155 grown in co-culture biofilms with AB6870155 significantly overexpressed aldehyde dehydrogenase, *ald1*, by ca. 40-fold and alcohol dehydrogenase, *adhP_2*, by 1.5-fold (Fig. S6) while decreasing expression of L-lactate dehydrogenase, *lldD_1*, by ca. 3-fold and L-lactate permease, *lldP*, by 2-fold (Supplementary Data 8). Increased expression of aldehyde dehydrogenase coincident with lower expression levels of lactate dehydrogenase has been shown to enhance ethanol production via glycerol fermentation in *K. pneumoniae* GEM167 mutant strains[32,33]. This suggests ethanol rather than its competing fermentation end-product, lactate (which is also provided as carbon source in SLMM), could be produced in the co-culture biofilms. Concordantly, *A. baumannii* AB6870155 co-cultures exhibited 3.2-fold increased transcript levels of alcohol dehydrogenase (NH10_00199) relative to pure cultures and no significant change in lactate transporters or dehydrogenase transcription levels (Supplementary Data 10), indicating probable utilisation of ethanol as a carbon source by AB6870155 in these co-culture biofilms. Oxidation of ethanol generates aldehyde via alcohol dehydrogenase, which requires effective removal due to its toxicity in high concentrations. To overcome this, *A. baumannii* utilising ethanol for assimilation converts the aldehyde into acetate via aldehyde dehydrogenase[34], which was found to have significantly higher transcript levels by ca. 3-fold (NH10_01161) in co-culture biofilms (Supplementary Data 10). Previous studies have shown that *A. baumannii* utilising ethanol as a carbon source also increases carbohydrate metabolism pathways including TCA cycle genes[34,35] which agrees with our findings (Fig. S4c, Supplementary Data 7).

### *K. pneumoniae* can cross-feed *A. baumannii*

To validate whether AB6870155 and KP6870155 do indeed interact synergistically at the metabolic level, we conducted cross-feeding experiments. The cells were grown in planktonic cultures, physically separated by a 0.2 μm filter (Fig. 3c) not permeable to cells, and with a single carbon source utilised exclusively by KP6870155: glycerol, mannose, galactose, serine, maltose and sucrose (Supplementary Data 9). For all carbon sources, AB6870155 growth occurred only in the presence of KP6870155 (Fig. 3d), signifying the requirement for KP6870155 generated metabolites, and thus cross-feeding. These results were confirmed using cross-feeding assays on minimal media agar plates supplemented with the selected carbon sources (Fig. S7). Inspection of the metabolic pathways present in both organisms suggested that KP6870155 could be producing ethanol as a fermentation end-product of metabolism, which may serve as a carbon source for AB6870155.

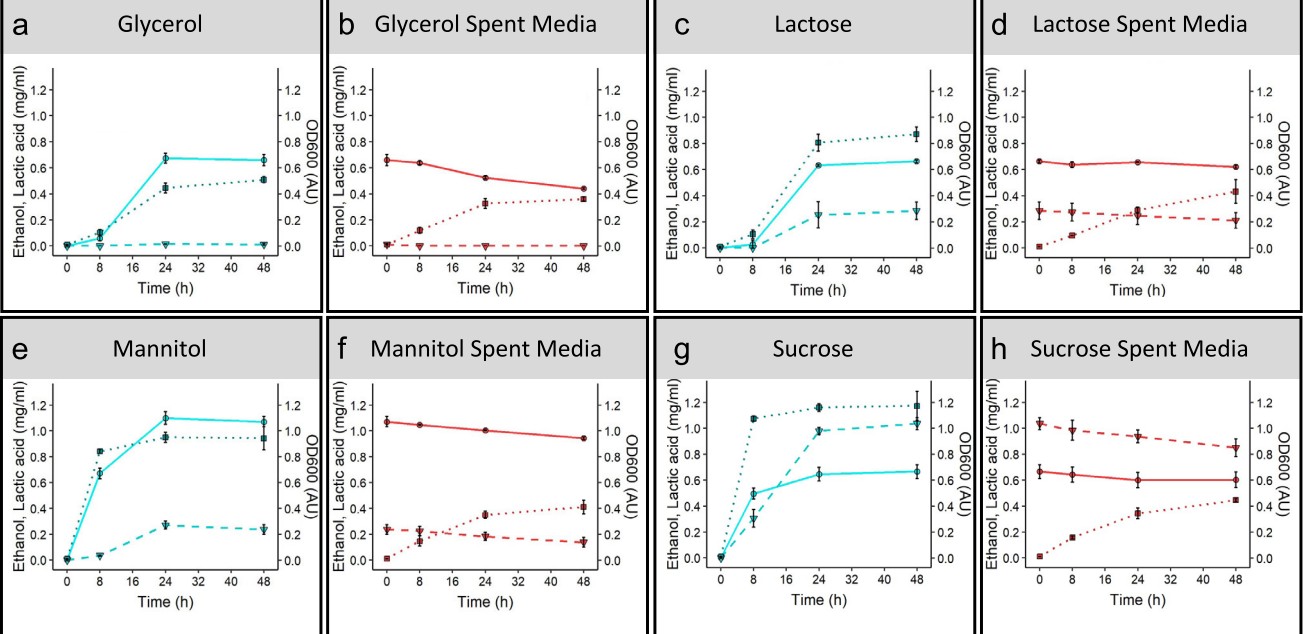

**Fig. 4 | Ethanol and lactate production by KP6870155 fed with various carbon sources and KP6870155 spent media utilisation by AB6870155.** HPLC measured ethanol and lactate production by *K. pneumoniae* KP6870155 (abbreviated as KP) grown with various carbon sources and consumption by *A. baumannii* AB6870155 (abbreviated as AB) grown in the KP6870155 spent media (*n* = 3 independent bacterial cultures). **a**, **b** glycerol, **c**, **d** lactose, **e**, **f** mannitol, **g**, **h** sucrose. Data are presented as mean values +/− SD. Source data are provided as a Source Data file.

We next tested *K. pneumoniae* KP6870155 for its ability to produce ethanol and lactate during metabolism of four carbon sources utilised by KP6870155 but not AB6870155: glycerol, lactose, mannitol and sucrose under biofilm conditions. We confirmed that KP6870155 was able to grow on these substrates (Fig. 4a, c, e, g) whereas AB6870155 did not grow on these substrates (Fig. S8). Ethanol and lactate were each produced by KP6870155 with most substrates provided, but the concentrations and ratios of these differed by substrate, with sucrose yielding the highest amount of secreted ethanol and lactate (Fig. 4g). The spent media of KP6870155 cultures grown in various carbon sources were then filter sterilized and inoculated with AB6870155 without any carbon source provided, except for the filtered spent media containing KP6870155 metabolites. Hence, the reduction in ethanol or lactate observed (Fig. 4b, d, f, h) is a result of AB6870155 consumption of these KP6870155 metabolic by-products. The consumption of these by AB6870155 is likely given the ability of *A. baumannii* to grow in L-lactic acid (Supplementary Data 9) and the previously reported ethanol utilization by *A. baumannii*[35]. A reduction in ethanol was observed during growth of AB6870155 in supernatants from glycerol and mannitol metabolism by KP6870155 (Fig. 4b, f). Glycerol metabolism by KP6870155 did not produce any lactate (Fig. 4a), thus it may imply that AB6870155 instead utilizes the ethanol produced from KP6870155 glycerol metabolism. Mannitol metabolism by *K. pneumoniae* resulted in significantly higher ethanol versus lactate production (Fig. 4e) and hence it is expected that AB6871055 would consume this abundant ethanol present in the mannitol fed KP6870155 supernatant. Sucrose fed KP6870155 produced the highest levels of lactate, and AB6870155 inoculated into sucrose fed KP6870155 metabolites also had the most active lactate consumption. These results indicate either lactate or ethanol generated by *K. pneumoniae* KP6870155 metabolism can be utilised by *A. baumannii* AB6870155 and that exchange of generated

metabolites is indeed possible, particularly when in proximity. This is not surprising as microbial cross-feeding often occurs within biofilms[36]. These data combined show that ethanol is a likely by-product of KP6870155 fermentation which is then assimilated by AB6870155 under these experimental conditions. Furthermore, this cross-feeding phenotype appears to also be widespread amongst other *A. baumannii* (AB5075, BAL062) and *K. pneumoniae* (ATCC 43816, NTUHK2044, SGH10) isolates tested and is not only seen for isolates specifically from co-infections (Fig. S8).

## Co-culturing induces changes to *A. baumannii* and *K. pneumoniae* biofilm formation, stress response and cell elongation

Since ethanol has previously been associated with increased *A. baumannii* virulence[34,35] through elevated stress response gene expression and enhanced biofilm formation, we next inspected the transcriptomics data for genes involved in biofilm formation and stress responses and noted significantly increased expression levels of these pathways for both species when co-grown in biofilms versus mono-cultured biofilms. Increased expression levels of stress response genes including those previously reported to be overexpressed in the presence of ethanol, namely chaperonin GroEL[35] and superoxide dismutase SodB was observed in *A. baumannii* AB6870155 co-cultured with *K. pneumoniae* KP6870155. Although both species displayed significant differential gene expression of stress-response genes in the presence of the other strain, they differed in which stress-response pathways were triggered (Supplementary Data 7). When grown in co-culture, both had significantly increased expression of oxidative stress associated genes, *sodB*, *sodC*, *oxyR* and *dnaJ*, the latter of which has been linked to ethanol stress response[37,38]. However, unlike the co-culture grown *A. baumannii*, where no significant differential expression occurred in universal stress proteins, *K. pneumoniae* had significantly increased expression of *uspB*,

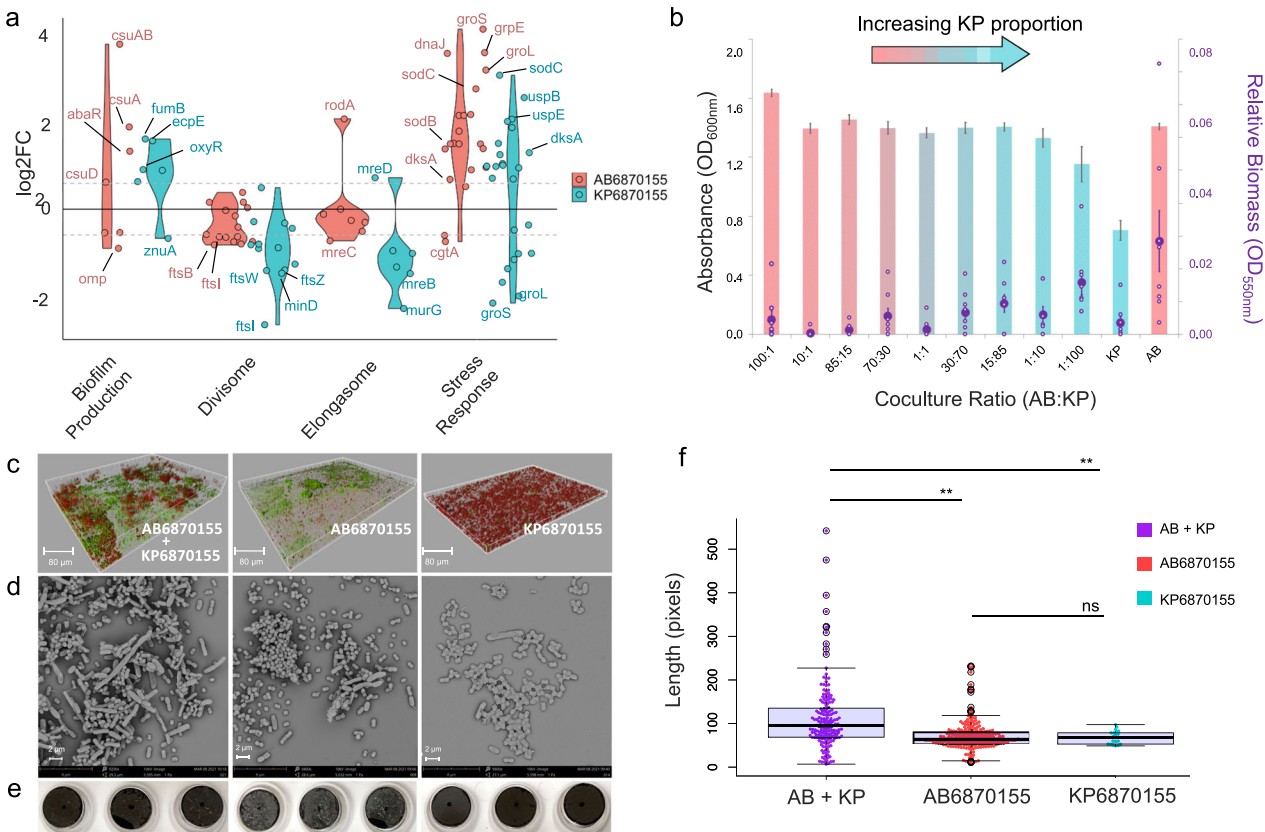

**Fig. 5 | Biofilm properties of *A. baumannii* AB6870155 and *K. pneumoniae* KP6870155 co-cultures. a** RNA-seq expression changes (log2FC) of AB6870155 and KP6870155 co-cultured biofilms relative to their respective mono-cultured biofilms. **b** Growth (bars) and biofilm formation (points) (measured as OD$_{550}$ after staining biofilms with crystal violet) of various ratios of AB6870155: KP6870155 grown planktonically; biofilm production detected by crystal violet assay ($n = 8$) and data are presented as mean values +/− SEM. **c** CLSM images of mono-culture and dual-species biofilms grown in flow-cell chambers for 3 days and stained using BacLight Live/Dead stain (ThermoFisher); green−live cells, red−dead cells. **d** SEM images of mono-culture and dual-species biofilms grown in microculture plates for 21 h ($n = 3$

independent bacterial cultures). **e** Physical appearance of single species or dual-species biofilms grown on coverslips for 21 h. **f** Mean cell length of pure versus co-culture biofilms grown on coverslips and imaged with SEM (AB + KP = AB6870155 + KP6870155 co-cultures); MicrobeJ software used to calculate cell lengths in SEM images. *P*-values were calculated using a two-sided Mann−Whitney test where significance values represent: \*\**p* ≤ 0.02, non-significant *p*-value = ns where *p* > 0.05. Boxes are bound by the first and third quartile with a horizontal line at the median and whiskers represent 1.5x the interquartile range. Dots represent individual cells ($n = 172$ AB + KP, $n = 225$ AB6870155, $n = 21$ KP6870155). Source data are provided as a Source Data file.

*uspE* and *uspG* (Fig. 5a). The role of universal stress proteins is multifaceted and somewhat elusive having roles in the stringent response, response to DNA damage and biofilm formation in various organisms[39]. Their role in *Klebsiella* response to multispecies interactions has not been reported prior to this study and warrants further investigation. Conversely, *A. baumannii* significantly overexpressed chaperone stress-response genes *dnaK, grpE, clpB, groL* and *groS*, while *K. pneumoniae* had decreased expression of these genes in co-cultures (Supplementary Data 7). DnaK/J, GrpE and ClpB salvage misfolded proteins during heat stress (36) and GroES/EL, GrpE and DnaK chaperones are important for protection against various other stresses including oxidative, osmotic and saline stress[37]. GroES/GroEL and DnaK/DnaJ/GrpE chaperone systems have also been reported to protect *E. coli* against aminoglycosides[40] thereby increasing its resistance to these antibiotics. Furthermore, a master regulator of stress, DksA, whose role in *A. baumannii* virulence was recently discovered[41], was also significantly overexpressed for both species in co-grown biofilms of AB6870155 and KP6870155.

Our data indicated that genes involved in biofilm formation in *A. baumannii* AB6870155 (*csuAB*) and *K. pneumoniae* KP6871055 (*fumB, ecpE, oxyR*) were significantly overexpressed when grown in co-cultures (Fig. 5a). To determine whether co-culturing resulted in enhanced biofilm formation, we performed crystal violet biofilm

assays to compare the biofilm biomass of mono-cultures with various ratios of co-cultures. *A. baumannii* mono-cultures showed higher biofilm formation compared to *K. pneumoniae* (Fig. 5b). Counterintuitively, co-cultures with higher proportions of *K. pneumoniae* exhibited thicker biofilms than those with higher proportions of *A. baumannii* (Fig. 5b), indicating that there may be distinct differences in biofilm architecture between mono- versus co-culture biofilms.

Given that various stresses in biofilms such as nutrient limitation, oxygen availability and antibiotics can lead to bacterial cell filamentation[42], we examined the differential expression profiles of cell shape-determining genes, particularly those involved in cell elongation and division[42,43], in co-culture versus mono-culture grown biofilms (Supplementary Data 7). Both co-culture grown AB6870155 and KP6870155 had lower expression of cell division genes, particularly those involved in Z-ring formation, which may indicate a decrease in cell division typically leading to filamentation[43]; these lower expression levels were more pronounced in KP6870155 (Fig. 5a). Interestingly, there were significantly increased transcript levels of *rodA* in *A. baumannii* co-cultures and no significant change in gene expression of other elongation-related genes that determine rod-shape morphology. Most elongasome genes had significantly lower transcript levels in *K. pneumoniae* grown in co-cultures which would result in shorter cell lengths.

**Table 2 | Minimum inhibitory concentrations for mono-cultures and co-cultures (1:1) of *A. baumannii* AB6870155 and *K. pneumoniae* KP6870155 (µg/mL)**

| Culture | Cefotaxime | Meropenem | Doxycycline | Gentamicin | Ampicillin |
|---|---|---|---|---|---|
| AB6870155 | 1024 | 0.25–0.5 | 8–16 | 512 | > 512 |
| KP6870155 | 16 | <0.125 | 2–4 | <1 | > 512 |
| A + K[a] | 1024 | 0.5 | 8 | > 512 | > 512 |

[a]A + K = AB6870155 + KP6870155.

Given the role of quorum sensing in biofilm formation[44], we investigated whether co-culturing versus mono-culturing led to any changes in these systems for both strains. Firstly, in AB6870155 we observed significantly increased expression of the *abaR* LuxR family transcriptional regulator gene (log2FC 1.3) (Fig. 5a), however, no significant change in expression of the *abaI* acyl-homoserine lactone (AHL) synthase gene was detected. Increased levels of the AbaR regulator may enable a greater level of activation of biofilm genes and other genes regulated by AbaR. *K. pneumoniae* has recently been shown to produce acyl-homoserine lactones (AHL)[45], however, no LuxI synthase homologue responsible for production of AHLs has been characterised thus far in *K. pneumoniae*. The KP6870155 strain contains a *luxR* gene and a *sdiA* gene that both encode LuxR-type transcriptional regulators, that may enable interspecies communication by responding to AHL signals[46]. KP6870155 did not show any significant changes to expression of these genes in response to being co-cultured with AB6870155. There was also no significant change in expression of its AI-2 system gene, *luxS*, encoding for an AI-2 synthase enzyme *S*-ribosylhomocysteine lyase (Supplementary Data 8). Hence, there was a lack of significant changes to *K. pneumoniae* quorum sensing in response to co-culturing with *A. baumannii* at the 24-h timepoint. These results demonstrate that both AB6870155 and KP6870155 appear to contribute to biofilm formation within co-cultures, but we could not see evidence of crosstalk between the quorum sensing systems in the two species, at least not during early stationary phases of co-culture growth.

To elucidate the architecture of mixed species biofilms of *A. baumannii* and *K. pneumoniae*, we visualised live/dead (ThermoFisher) stained dual-species and mono-culture biofilms with confocal laser scanning microscopy (CLSM) in a flow-cell continuous system. Interestingly, three-day old *K. pneumoniae* mono-culture biofilms comprised mostly of dead or cell membrane permeable cells, while cells in *A. baumannii* biofilms were mostly live (Fig. 5c). Co-grown biofilms displayed a mixed population of live and dead cells in various niches of the biofilm that resembled the overall biofilm structure of respective mono-culture biofilms (Fig. 5c). This might suggest the presence of both populations of cells in dual-species biofilms, enabling for efficient transfer and sharing of metabolites. However, co-culturing could also have influenced the cell viability.

Visual inspection of biofilms grown on the coverslips suggested a more pronounced biofilm biomass for *A. baumannii* AB6870155 compared to *K. pneumoniae* KP6870155 mono-culture biofilms (Fig. 5e). To get a closer look, we imaged the dual-species and mono-culture biofilms grown in SLMM via scanning electron microscopy (SEM). Strikingly, SEM suggested an increased subpopulation of elongated cells in co-culture biofilms (Fig. 5d) that was determined to be statistically significant using MicrobeJ software[47] to calculate average cell lengths (Fig. 5f). Although filamentation of *A. baumannii* and *K. pneumoniae* has been previously observed in cells exposed to various classes of antibiotics[48–51], this phenomenon in response to mixed-species co-culturing has not yet been reported for bacteria[42]. The previously observed filamentation of these species in response to antibiotic stress corresponds with the increased expression of stress response genes observed for both species in the co-culture grown biofilms (Fig. 5a).

## Cefotaxime cross-protection of *K. pneumoniae* by *A. baumannii*

Biofilms facilitate a complex array of intra- and interspecies interactions enabling signalling pathway interactions, exchange of genetic material and nutrients through cross-feeding, that can alter antibiotic resistance in microbial communities[52], and even cross-resistance to antimicrobials[53–55]. Hence, we tested whether the co-infection isolates exhibited any cross-protection against antibiotics during biofilm conditions. We first established the MICs of both mono and co-cultures (1:1 ratio) of AB6870155 and KP6870155 to various classes of antibiotics including gentamicin (aminoglycoside), tetracycline (glycylcycline), and several β-lactams including ampicillin (penicillin), meropenem (carbapenem) and cefotaxime (3rd generation cephalosporin). AB6870155 mono-culture was highly resistant to cefotaxime, ampicillin, and gentamicin while KP6870155 was highly resistant to ampicillin and moderately resistant to cefotaxime (Table 2). Both were susceptible to meropenem according to EUCAST breakpoints (Table 2). Cefotaxime and gentamicin were the most likely antibiotics for which cross-protection may occur given that one strain had a significantly higher MIC than the other. Looking at the expression profiles for antibiotic resistance genes, we observed that plasmid encoded *bla_2* SHV-12 type cephalosporinase, had significantly lower expression in KP6870155 (log2FC −2.4) while Tn*6168*-bound *ampC* transcription was not significantly lower in AB6870155 (Fig. S11). Given that cross-protection is well documented for β-lactams[56–58] but not aminoglycosides[59], we investigated whether the cross-protection phenomenon occurred in co-cultures exposed to cefotaxime.

Since *A. baumannii* biofilm formation can significantly enhance cefotaxime resistance[60], we tested whether it could provide cross-protection to *K. pneumoniae*. We assayed for growth and biofilm formation using crystal violet staining for AB6870155 to KP6870155 in the presence and absence of cefotaxime (at 512 µg/mL). Because different relative abundances can influence interspecies interactions[61], we tested various co-culture ratios of AB6870155 to KP6870155. As expected from the MICs for these strains (Table 2), cefotaxime at 512 µg/mL killed KP6870155 but not AB6870155, however growth did not significantly decrease for co-cultures with higher proportions of KP6870155 (Fig. S9a). Given the variations in growth between the mono-cultures and various co-cultures, we normalized biofilm biomass (measured by crystal violet staining at $OD_{550nm}$) against culture growth (measured at $OD_{600nm}$) to calculate the biofilm production relative to growth (Fig. S9b). For co-cultures with higher proportions of AB6870155, there was a significant increase in biofilm production relative to culture growth for those grown in the presence of cefotaxime, particularly for the 70:30 ratio of AB6870155:KP6870155 (Fig S8b).

To determine whether any *K. pneumoniae* KP6870155 survived exposure to the cefotaxime concentration 32x its MIC, we plated 512 µg/mL cefotaxime treated co-cultures and mono-cultures onto differential agar, allowing for distinction between *K. pneumoniae* β-galactosidase producing cells (red colonies) and *A. baumannii* β-galactosidase deficient cells (white colonies). This revealed that cefotaxime-treated co-cultures, particularly those with a higher abundance of *K. pneumoniae*, had a surviving population of KP6870155 (Fig. 6a). To further test for cross-protection in cefotaxime, we exposed mono- and co-cultures of AB6870155 and KP6870155 to 512 µg/ml cefotaxime and plated the cultures onto MacConkey agar at

time zero and 21 h post exposure to see whether a heteroresistant population could be revived. However, no colonies grew on the Mac-Conkey agar plates from *K. pneumoniae* mono-cultures treated with 512 μg/mL cefotaxime after 21 h of exposure (Fig. S10a), and hence the surviving KP6870155 population within the co-cultures were likely not heteroresistant cells. Further, we re-isolated a representative KP6870155 from AB6870155 co-cultures grown in 512 μg/ml cefotaxime and found that its MIC to cefotaxime had decreased 2-fold, from 16 μg/ml to 8 μg/ml (Fig. 6a). To further elucidate the mode of cross-protection, we performed a Millicell hanging insert experiment where KP6870155 was physically separated from AB6870155 but could share the same media and any secreted molecules therein. Growth of KP6870155 was significantly higher when sharing media with AB6870155 than when grown alone (Fig. 6b) in the presence of cefotaxime (CEF) treatment but not when using the beta-lactamase inhibitor, sulbactam, in combination with cefotaxime (CEF:SUL). Cephalosporinase secretion by *A. baumannii* was diminished in the CEF:SUL combined treatment (Fig. S10b) indicating cross-protection of KP6870155 by AB6870155 is via secreted cephalosporinases.

## Virulence and drug resistance gene expression in *K. pneumoniae* and *A. baumannii* co-cultures

We next explored the expression patterns of various virulence-related genes in *A. baumannii* and *K. pneumoniae* co-grown biofilms in SLMM, including genes involved in antibiotic resistance, secretion systems, iron-acquisition, polyamine synthesis and the phenylacetate (PAA) pathway. Overall, most secretion system pathways had decreased expression levels in AB6870155 in co-culture with KP6870155 and no significant changes to these genes transcription were observed in KP6870155 except for significantly decreased transcript levels of *virB4* gene encoding a Type 4 secretion system (T4SS) protein. The type VI secretion system (T6SS) is a multi-component machinery that is assembled when *A. baumannii* is triggered to compete with antagonising Gram-negative bacteria and while not normally required for virulence[13], its inherent role in bacterial competition should have an impact on polymicrobial infections. This weapon-like machinery is energetically costly and *A. baumannii* has control mechanisms in place to regulate its expression[62]. Some strains have a large conjugative multidrug resistance plasmid encoding repressors that negatively regulate T6SS expression[63], while other strains harbor a single L749R amino acid substitution in the *vgrG* gene, encoding for a key T6SS protein, that acts as an inhibitor of T6SS with this substitution present[62]. Since both *vgrG* (NH10_02760 and NH10_02562) transcripts, found in regions of genome plasticity (Supplementary Data 1), were significantly higher in AB6870155 co-cultured with KP6870155 while most other T6SS genes, *tssB, tssF, tssC, tssE* and tube protein gene *hcp* (NH10_02523) had significantly lower transcript levels (Fig. S15, Supplementary Data 10), it appears that both *vgrG* genes could have a role in inhibition of T6SS. We aligned AB6870155 *vgrG* genes with those from other *A. baumannii* strains but did not find the L749R amino acid substitution. AB6870155 harbors two *vgrG* genes which are highly conserved with A85 *vgrG*, sharing 100% amino acid identity. They are also highly similar to ACICU *vgrG* genes but more distantly related to ATCC 17978 vgrG2 (Fig. S14). The majority of amino acid differences between AB6870155 and ACICU *vgrG* genes occurred in the C-terminal region of the protein which has been demonstrated to be essential for T6SS assembly[62]. Further investigation into the AB6870155 *vgrG* genes is warranted to determine whether they play a role in T6SS inhibition. The decreased expression levels of most T6SS genes in co-grown biofilms with KP6870155 indicate a lack of T6SS-based competition between these two bacteria.

Despite the syntrophic interactions between these pathogens, their overexpression of stress genes (Fig. 5a) in co-cultures indicates some level of tension to their dynamic. Significant overexpression of specific antibiotic resistance genes was observed in both AB6870155 and KP6870155 when co-grown in biofilms relative to their pure cultures

(Fig. S11). *K. pneumoniae* had significantly increased expression of *emrE* multidrug transporter (log2FC 3.6) and streptomycin 3″-adenylyll-transferase, *ant1* (log2FC 2.3), that confers antibiotic resistance to the aminoglycoside streptomycin. Similarly, *A. baumannii* had a 3.5-fold increased expression of *aadA* (Fig. S11), an aminoglycoside nucleotidyltransferase ANT(3″)-IIa which is located on RGP3 (Supplementary Data 1), shares 89.4% nucleotide identity with KP6870155 *ant1* as determined by Needleman-Wuncsch pairwise alignment in ClustalW[64], and also confers resistance to streptomycin. AB6870155 also had moderate 1.6-fold overexpression of *aacC1*, gentamicin 3-N-acetyltransferase (Supplementary Data 10). Neither of these bacteria are known to be capable of producing aminoglycoside antibiotics and thus the increased expression of these resistance genes during mixed-species growth points to a possible alternative role as has been observed in *Mycobacterium tuberculosis*, where the aaC(2′) enzyme also functions in acetylation of peptidoglycan[65]. The AB6870155 antibiotic resistance genes with the most decreased transcript levels in co-cultures were *adeI, adeJ, adeA* and moderately lower expression (−1.6-fold) of *adeK*. AdeIJK is a clinically important resistance-nodulation-division (RND) antibiotic efflux system that confers resistance to a broad range of antibiotics[66], has a role biocide resistance[67] and is involved in maintenance of lipid homeostasis likely through fatty acid export[68]. *K. pneumoniae* KP6870155 grown in co-cultures with AB6870155 had significantly decreased expression of *ftsI, bla_2, msbA, acrA* and *mdtJ* (Fig. S11, Supplementary Data 8). The *ftsI* and *bla*$_{SHV}$ genes encode for peptidoglycan D,D-transpeptidase and β-lactamase SHV-12 proteins respectively, which enable resistance to β-lactam antibiotics and cephalosporins. The lower transcript levels of these genes in the co-cultures indicates a potential reliance on AB6870155 β-lactamases.

Interestingly, AB6870155 grown in co-culture biofilms showed an increased expression of *amvA* which encodes for a major-facilitator superfamily (MFS) multidrug efflux pump whose natural substrates include the polyamines, spermidine and spermine[69]. Additionally, transcript levels of the *aceI* gene in AB6870155, encoding the AceI efflux pump, were also significantly higher (log2FC 5.5) in the co-cultured cells compared to those in mono-culture. AceI was originally identified as a pump capable of conferring resistance to the biocide chlorhexidine[70], but has since been found to transport polyamines, including putrescine and cadaverine[71]. Its high-level expression in mixed-species biofilms along with increased expression of *amvA* could reflect a higher concentration of polyamines in the co-culture environment. This hypothesis is further supported by an increase, ca. 2-fold, in transcript abundance observed for the SapB putrescene exporter in KP6870155. Additionally, *A. baumannii* AB6870155 co-cultured with KP6870155 had 13-fold increased expression of the *gabT* gene encoding for 4-aminobutyrate-2-oxoglutarate transaminase which has been proposed to function in putrescene homeostasis[72]. Polyamines have multiple functions in bacteria including biofilm formation, virulence and siderophore synthesis[73,74]. Strikingly, genes with the most dramatic decrease in transcript levels, 30–60-fold, in *A. baumannii* co-cultured with *K. pneumoniae* belonged to the *bas* (*A. baumannii* acinetobactin biosynthesis) genes responsible for biosynthesis and transport of the acinetobactin siderophores, the *bau* (*A. baumannii* acinetobactin utilisation) genes and the *bar* (*A. baumannii* acinetobactin release) genes[75]. These genes form a cluster of operons in the AB6870155 genome (Fig. S12b). Additionally, the iron uptake regulators, *fecI* and *fecR* located in RGP16 with *hemO*, a heme oxygenase, along with several hypothetical genes also had significantly lower expression levels (Fig. S13, Supplementary Data 1). Despite the decreased expression of iron acquisition genes, the iron-dependant Class I fumarate hydratase gene, *fumA*, had nearly 3-fold higher expression while the iron-independent Class II fumarase, *fumC*, had 2-fold lower expression which directly contrasts to what has been observed for *A. baumannii* grown under iron limiting conditions[76]. Hence the co-cultures reflected transcription patterns expected during iron replete conditions,

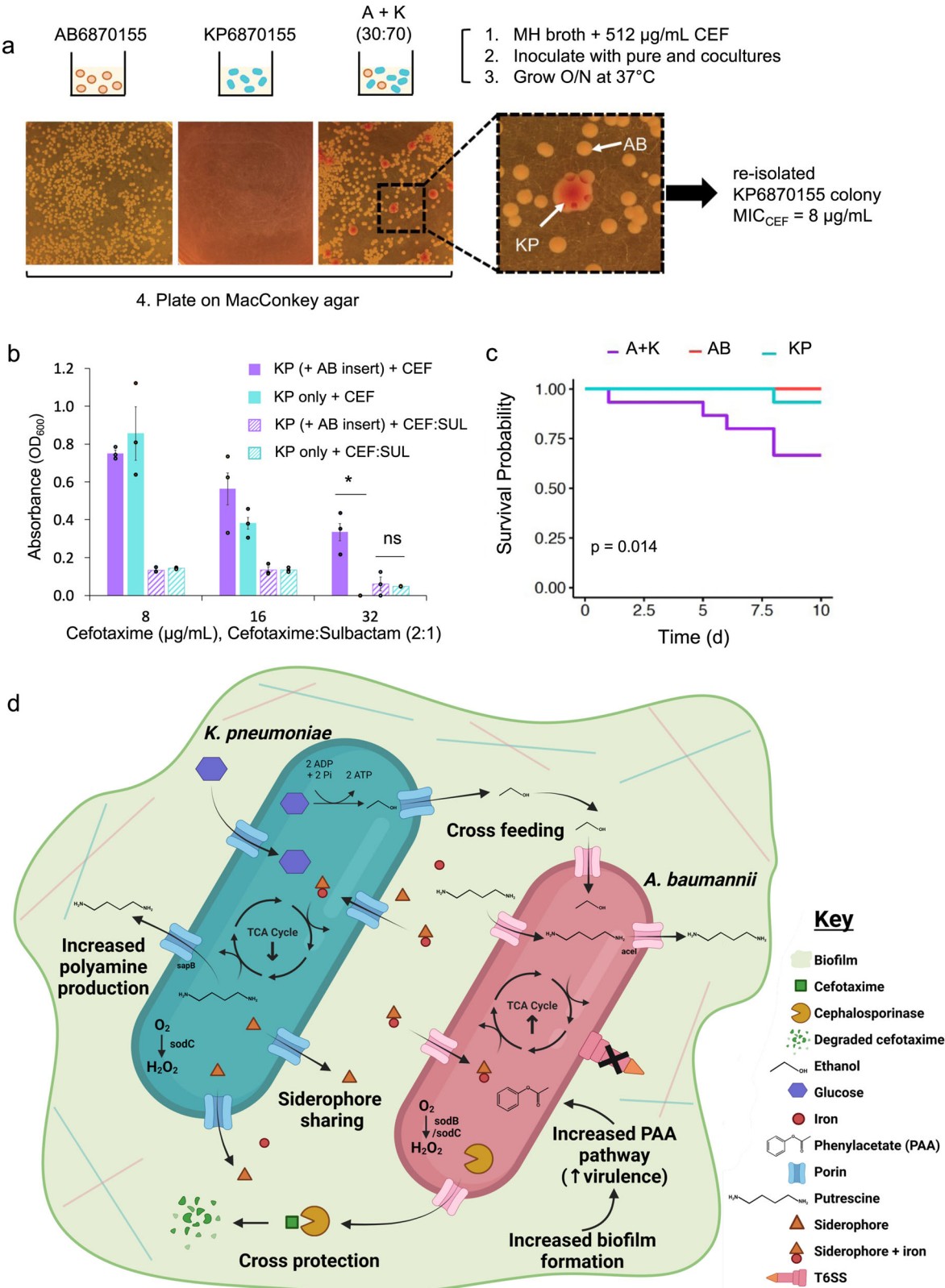

despite the presence of *K. pneumoniae* as an additional competitor for iron. Besides iron acquisition genes found in the major siderophore operons, bacteria also contain siderophore receptor genes distributed more randomly in the genome that can recognize exogenously produced siderophores[77]. In line with this, a TonB-dependent siderophore receptor gene, NH10_03345, had over 2-fold increased expression in co-culture grown *A. baumannii* and its ability to uptake

xenosiderophores warrants further investigation as TonB-dependent receptors are known to transport xenosiderophores in many Gram-negatives including *A.baumannii*[78]. Interestingly, siderophore production in KP6870155 was not as drastically changed for most genes, with *fepA_2*, a ferrienterobactin receptor, having 2-fold increased expression (Fig. S12a). This suggests a sharing of *K. pneumoniae* siderophores with *A. baumannii* as is possible with secreted

**Fig. 6 | Effects of co-cultures on antibiotic resistance and virulence. a** Antibiotic cross-protection assay in 512 μg/mL cefotaxime. A + K denotes AB6870155 and KP6870155 co-cultures at a 30:70 ratio respectively. Inset of MacConkey agar plate shows KP (KP6870155) colonies appearing red due to *K. pneumoniae* β-galactosidase activity; AB (AB6870155) lacks this enzyme and appears as white colonies. **b** Growth of *K. pneumoniae* KP6870155 in cefotaxime only (+ CEF) and cefotaxime + sulbactam (+ CEF:SUL at 2:1) as measured by optical density (OD$_{600}$). "KP only" represents *K. pneumoniae* KP6870155 grown in wells without sharing media with *A. baumannii* AB6870155 while "KP (+ AB insert)" represents KP6870155 physically separated from AB6870155 via a Millicell hanging insert and able to share media ($n = 3$). Data are presented as mean values +/− SEM. Significance was measured by paired Student's *t*-test (two-tailed) where *represents a $p < 0.05$ ($p = 0.03$) and ns (not significant) represents a $p > 0.05$. **c** Kaplan–Meier curves of single injected AB6870155 (AB) and KP6870155 (KP) and co-injected (A + K) at a 1:1 ratio in *G. mellonella* (*p*-value based on log-rank test); PBS only controls had 100% survival (overlap with AB curve). **d** schematic representation created with BioRender and ChemDraw (v22.0.0) of *A. baumannii* AB6870155 and *K. pneumoniae* KP6870155 interactions. Source data are provided as a Source Data file.

siderophores[79]. Finally, there was a divergence in *paa* operon gene expression, with AB68970155 having significantly increased expression and KP6870155 having significantly decreased expression of the phenylacetic acid (PA) catabolic pathway. This pathway plays a role in aromatic compound metabolism and in *A. baumannii* virulence[80].

### Coinfection with *A. baumannii* and *K. pneumoniae* increases in vivo virulence

To gain an understanding of how co-culture dynamics affects virulence in the host during infection, we utilised the wax moth larvae *Galleria mellonella* as an in vivo infection model. *G. mellonella* was chosen as an infection model given its suitability for studying bacterial co-infections[81]. $1 \times 10^5$ cells of *K. pneumoniae* KP6870155 and *A. baumannii* AB6870155 were injected individually and mixed in equal proportions (with the total number of each species kept constant at $10^5$ cells each) just prior to injection and larvae survival over a 10-day period was monitored. Co-infections with the two strains had the highest level of virulence, killing 33% of larvae by day 10, single infection with *K. pneumoniae* was the next most virulent with 7% of larvae dead by day 10, whereas *A. baumannii* AB6870155 was least virulent, with a 100% survival rate (Fig. 6c). Hence the combination of the two strains at the injection dose tested appeared to have a synergistic killing effect where the total larvae dead significantly exceeded that of both strains injected as single infections, however further characterisation is necessary to distinguish whether the effect is synergistic versus additive. This effect was also observed, albeit to a lesser extent, when the inoculum size of single infection matched that of coinfection (Fig. S16).

## Discussion

Polymicrobial infections are prevalent in human diseases including cystic fibrosis, urinary tract infections, pneumonia, wound infections and inflammatory bowel disease and can make antimicrobial therapies less effective and increase severity of disease[82–84]. Multi-species infections are made possible through synergistic interactions between co-infecting pathogens and the mechanisms of these interactions include cross-feeding, communication through chemical signalling, direct contact through biofilms and cross-protection against environmental stress[4,52,82,85]. In this study, we found that cross-feeding, biofilm production and cross-protection were mechanisms of synergy between *A. baumannii* AB6870155 and *K. pneumoniae* KP6870155. Such symbiotic interactions, while friendly from a bacterial perspective can wreak havoc in the infected host in the context of polymicrobial infections.

Ramsey et al.[86] demonstrated that the in vivo persistence of the pathogen *Aggregatibacter actinomycetemcomitans*, was dependent on being cross-fed L-lactate by co-infecting pathogen *Streptococcus gordonii*. Through cross-feeding and successional co-feeding experiments, we found that *A. baumannii* AB6870155 was able to consume *K. pneumoniae* KP6870155 fermentation by-products, ethanol and lactate. The biofilm matrix can serve as a medium for more efficient exchange of metabolites through cross-feeding[87]. The co-cultures in our study exhibited increased levels of biofilm production with increasing proportions of *K. pneumoniae*. Although it is not clear why higher proportions of *K. pneumoniae* increased biofilm formation, a possibility

could be secretion of biofilm production inducing signals within the co-cultures. This phenomenon has been observed by Keogh et al. where *Enterococcus faecalis* secretes L-ornithine which acts as a signal to promote biofilm production in its co-culture partner, *E. coli*[88]. Their study also found that siderophore production by *E. coli* was enhanced by the presence of *E. faecalis*. Interestingly, genes with the most decreased expression levels in *A. baumannii* grown in co-cultures with *K. pneumoniae* were those encoding siderophores, while *K. pneumoniae* did not experience such a drastic decline in siderophore expression, indicating potential siderophore sharing. This finding agrees well with the eco-evolutionary model for siderophore production that predicts downregulation of siderophores when they are used as a public good rather than when being privatized[89]. This sharing would mitigate any stealing of iron between interspecies produced siderophores while allowing *A. baumannii* to trade-off the energetically costly siderophore production for allocation of energy towards growth. Some of the iron acquisition pathway genes with lower expression levels in *A. baumannii* including iron acquisition regulators, *fecI* and *fecR*, and heme oxygenase, *hemO*, were located within an RGP containing many hypothetical proteins that also had significantly lower transcript levels (Fig. S13). This observation may indicate a recently acquired function for this *A. baumannii* strain and nearest relative strains that appears to have a role in mixed-species interactions with other coinfecting pathogens, however further investigation is warranted.

Previous work by Adamowicz and colleagues, demonstrated that polymicrobial communities engineered towards obligate mutualism through cross-feeding, have antibiotic susceptibilities similar that of the "weakest link" in the community. This means the polymicrobial MIC is the same as that of the microbe with the lowest MIC[52], however the MICs were measured in defined media. In contrast, our findings showed that the MICs of co-cultures grown in complex media (MH), where carbon sources are accessible to both species, were higher or the same as that of the more resistant strain, *A. baumannii* AB6870155. Adamowicz and colleagues do explain that exceptions to this "weakest link" rule exist, when one of the community members either excretes an antibiotic degrading compound into the environment or activates tolerance mechanisms in its neighbours, thereby providing antibiotic cross-protection[52]. Liao and colleagues[90] found that cross-protection of various carbapenem susceptible strains including *K. pneumoniae*, *P. aeruginosa*, *E. coli* and *Enterobacter cloacae* was facilitated by carbapenemase-producing *A. baumannii*, impeding efficacy of antibiotic treatment. Similarly, Smith et al. found that co-culturing *Staphylococcus aureus* with carbapenem resistant *A. baumannii*[91] significantly reduced the bactericidal activity of the antibiotic meropenem against *S. aureus*. In this work, we observed that *A. baumannii* AB6870155 was able to cross-protect *K. pneumoniae* KP6870155 against inhibitory concentrations of cefotaxime that would normally kill this *K. pneumoniae* strain. However, in the case where *A. baumannii* is more susceptible to cefotaxime, the MIC of KP6870155 and *A. baumannii* cocultures is lower than that of KP6870155 pure cultures (Supplementary Data 12), indicating there is a fine balance between which strain provides cross-protection and how much each strain is benefitting from the interaction. These previous studies and the work here demonstrate a propensity for *A. baumannii* to shelter rather than kill certain co-habiting pathogens.

This syntrophic behaviour from *A. baumannii* requires certain competitive traits to be repressed. *A. baumannii* is equipped with a tightly regulated T6SS that when active, can effectively kill competing Gram-negative and Gram-positive bacteria, including killing strains within the same species[92,93]. However, in certain *A. baumannii* strains, T6SS expression is suppressed by either negative regulators harboured on a multi-drug resistant plasmid[63] or through a variant of the VgrG spike protein[62]. The *A. baumannii* AB6870155 strain described in this study, decreased expression of T6SS when co-cultured with *K. pneumoniae* KP6870155. This disarmament of the T6SS was not as pronounced from the perspective of KP6870155, however the majority of predicted T6SS genes in KP6870155 (Supplementary Data 8) did not have significantly changed expression within the co-cultures. Hence it appears that synergistic interactions between the strains tips the balance toward co-operation rather than interspecies bacterial warfare, even though both bacteria demonstrated significantly higher transcript levels of various stress response genes when co-grown in biofilms. Stress responses in bacteria can lead to virulence[94] and we thus studied whether co-infections with these two strains increased virulence in vivo. We observed a significant decrease in survival of *G. mellonella* co-infected with *K. pneumoniae* KP6870155 and *A. baumannii* AB6870155 as compared to single strain infections, indicating that coinfection with *K. pneumoniae* and *A. baumannii* can lead to increased virulence in vivo.

The symbiotic interactions observed here, whereby *K. pneumoniae* can provide *A. baumannii* with metabolites in exchange for shelter from environmental stress (Fig. 6e) is reminiscent of lichen-style symbioses and cements our understanding of why we find these pathogens co-isolated from polymicrobial infections in clinical settings. This work demonstrates the need to identify polymicrobial infections and test coinfecting pathogens for cross-protection prior to antibiotic administration. This will ensure the most effective regimen can be assigned that has minimal likelihood for recalcitrance to therapy caused by cross-protection between coinfecting pathogens.

## Methods

### Bacterial strains and growth conditions

AB6870155 and KP6870155 strains were previously co- isolated from the sputum of a patient with a lung infection[1] and are available upon request. All cell culture incubations of these isolates were performed at 37 °C. Unless otherwise stated, for all the experiments performed in this study, cells were collected in mid-log phase after being sub-cultured from overnight cell cultures grown aerobically at 37 °C with shaking at 200 rpm. Mueller-Hinton (MH) medium, a rich medium routinely used for antimicrobial susceptibility testing in laboratories worldwide that contains starch, casein acid hydrolysate and beef extract rich in amino acids, synthetic lung mimicking medium (SLMM)[95], which is a more defined medium comprised of glucose, lactate and amino acids as available carbon sources (22), and M9 minimal medium were used for cell growth as indicated.

### Antibiotic susceptibility and cross-protection testing

Antibiotic susceptibility testing was done in Mueller-Hinton (MH) cation adjusted medium using the broth microdilution method as previously described[96]. Briefly, cultures were streaked from frozen glycerol stocks onto MH agar plates and incubated at 37 °C overnight. Five colonies were picked and normalized in MH broth to $1 \times 10^6$ CFU/ml. For co-culture tests, *A. baumannii* AB6870155 and *K. pneumoniae* KP6870155 were mixed at the defined ratios tested. Pure cultures and co-cultures were added to 2-fold serial dilutions of the antibiotics tested giving a final inoculum size of $5 \times 10^5$ CFU/ml. The inoculated 96-well plates were incubated at 37 °C for 18 h and cell growth was measured at $OD_{600}$ with a spectrophotometer. To examine *A. baumannii* AB6870155 cross-protection of *K. pneumoniae* KP6870155 when the two species have physical contact, pure and co-cultures were exposed to 512 µg/ml

cefotaxime and their survival was measured via colony counts of spread plates at time 0 (upon initial exposure to cefotaxime) and time 21 h following 21 h of static incubation at 37 °C. Cultures were diluted in MH broth to $1 \times 10^3$ CFU/ml and spread plated onto MacConkey agar, allowing for differentiation between *K. pneumoniae* (colonies appear red due to β-galactosidase activity) and *A. baumannii* (colonies appear white due to lack of β-galactosidase). Cross-protection experiments were also performed using Millicell hanging inserts (Millipore) where *A. baumannii* AB6870155 was inoculated into inserts that were placed into wells of a 24-well plate inoculated with *K. pneumoniae* KP6870155 at $5 \times 10^5$ CFU/ml. Various concentrations (8–32 µg/ml) of cefotaxime and cefotaxime:sulbactam (2:1 ratios) were tested and cell growth was measured at $OD_{600}$ with a spectrophotometer after 21 h incubation at 37 °C. To test whether *A. baumannii* cephalosporinases were secreted, cell pellet versus extracellular secreted fractions were prepared and applied to nitrocefin discs (Merck), where appearance of a red colour indicates presence of cephalosporinases. *A. baumannii* grown in the presence of cefotaxime and cefotaxime:sulbactam, normalised to the same $OD_{600}$ was pelleted at 3000 g for 10 min. Supernatants were filter sterilised with a 0.2 µm filter (cell filtrate fraction) and cell pellets were washed twice in 1x PBS with centrifugation of 3000 g for 10 min between washes (cell pellet fraction). A 10 µl drop of cell filtrate and washed cell pellet was placed onto nitrocefin discs and colour change was recorded after 5 min of incubation at room temperature.

### Determination of co-culture proportions by species-specific qPCR

Single colonies of *A. baumannii* AB6870155 and *K. pneumoniae* KP6870155 were separately inoculated into MH or SLMM and grown aerobically at 37 °C overnight. Each strain was sub-cultured into fresh media and grown to mid-exponential phase ($OD_{600}$ 0.3 to 0.7). This sub-culture was split across 3 replicates for each condition, growth in MH broth and growth in SLMM. Co-cultures were inoculated in 20 ml fresh media at ~$5 \times 10^5$ CFU/ml total (~$2.5 \times 10^5$ CFU/ml of each strain) and grown at 37 °C, 120 rpm. At each timepoint, 1–2 ml samples were taken, spun down at 10,000 x *g* for 10 min and the supernatant discarded. Cell pellets were frozen at −30 °C until gDNA extractions were performed.

Cell pellets were thawed, re-centrifuged at 10,000 x *g*, 1 min, and any remaining media removed with a pipette. The gDNA was extracted using a DNeasy UltraClean Microbial Kit (Qiagen) as per the manufacturer's instructions with the following modifications: samples were incubated at 70 °C for 10 min after addition of solution SL and for denser cultures (above $OD_{600}$ ~0.8) 1 ml of input culture was used. Extracted gDNA samples were quantified using the Qubit dsDNA Broad Range assay in a Qubit 4 Fluorometer (Invitrogen) and all samples diluted to 5 ng/µl. The qPCR was performed with the QuantiNova SYBR Green PCR Kit (Qiagen) as per the manufacturer's instructions with a 10 µl total reaction volume and 5 ng of gDNA template. Primers (Integrated DNA Technologies Inc.) specific to AB6870155 targeted the $bla_{Oxa-53}$ gene (AB_oxa-53), forward sequence 5′-GGAAGTGAAGCGTGTTGGTT-3′ and reverse sequence 5′-CAAACTGTGCCTCTTGCTGA-3′. Primers (Integrated DNA Technologies Inc.) specific to KP6870155 targeted *acrB* (KP_acrB), forward sequence 5′-GTCGATTCCGTTCTCGGTTA-3′ and reverse sequence 5′-GCAGACCCACCTGGAAGTAA-3′. Thermocycling conditions were as follows: initial 95 °C 2 min, cycling 95 °C for 5 s then 60 °C for 10 s for 40 cycles. Primer efficiencies were 97.83 %, R2 = 0.999 for AB_oxa-53 and 97.92 %, R2 = 1.000 for KP_acrB. Co-culture ratios were calculated with the formula in equation (1):

$$(1)\quad \text{AB : KP ratio} = 2^{-(AB_{Cq} - KP_{Cq})}$$

Where: $ABc_q$ is the average $C_q$ value for AB and $KPc_q$ is the average $C_q$ value for KP at each time point. Proportions of each strain in the coculture were calculated by converting the ratio to a fraction of 1.

## DNA sequencing and genome analysis

DNA was isolated from *K. pneumoniae* and *A. baumannii* overnight cultures that were taken from colonies grown on CHROMagar plates (to confirm pure cultures) with the Qiagen DNeasy UltraClean Microbial Kit. Oxford Nanopore long-read sequencing and Illumina sequencing on the NextSeq 550 platform with the Nextera kit were performed at the Microbial Genome Sequencing Center (MiGS). Quality of the sequence data was assessed with FastQC v 0.11.9 (available from http://www.bioinformatics.bbsrc.ac.uk/projects/fastqc). Nanopore reads were quality filtered with Filtlong v 0.2.0 with min_length parameter set to 1000, keep_percent set to 90 and target_bases set to 500000000[97] and Illumina reads were quality filtered with nextseq-trim = 20 parameter and adapters were trimmed using cutadapt v 1.18 using parameters -a CTGTCTCTTATACACATCT -A CTGTCTCTTATACACATCT and minimum-length 36:36 (Martin, 2011, https://doi.org/10.14806/ej.17.1.200). The sequence reads were hybrid assembled into complete genome sequences using Unicycler v 0.4.9b[98] (with N50 values equalling the size of each strain's chromosome) and assembly graphs were visualised with Bandage[99]. Annotation was performed with Prokka 1.14.6 using the -rfam parameter to find ncRNAs[100]. The genome sequences were deposited in the GenBank database (BioProject ID PRJNA263680; accession number JRWO00000000 v02 for AB6870155, TBA for KP6870155). To find putative functions for proteins annotated as hypothetical, the eggNOG-mapper tool[101,102] was used. The complete genomes of 172 *A. baumannii* strains were downloaded for constructing the *A. baumannii* pangenome and 530 complete genomes from GenBank were downloaded for constructing the *K. pneumoniae* pangenomes (Supplementary Data 3–4). The pangenomes were constructed using PPanGGOLiN[31] v 1.1.108 using the panrgp command to find regions of plasticity. The SnapGene® software v 3.1.4 (from insightful Science; available at snapgene.com) was used to generate plasmid maps and PlasmidFinder was used to determine incompatibility groups[27]. FastANI[103] was used to calculate the average nucleotide identity (ANI) and alignment fractions (AF) of the 172 sequenced *A. baumannii* genomes and 530 assembled *K. pneumoniae* genomes. A distance matrix was generated with the parseDistanceMatrix function in R from the FastANI output. The ward.D2 method in the hclust R function was applied to the distance matrix and the output was converted to a dendrogram using the as.phylo() function using the R ape (v5.5) package[104]. For identification of antimicrobial resistance genes in the assembled *A. baumannii* AB6870155 and *K. pneumoniae* KP6870155 genomes, the Comprehensive Antibiotic Resistance Database (CARD)[105] v.2.0.3 was queried. Genomes were submitted to the Transporter Automated Analysis Pipeline (TransAAP; www.membranetransport.org)[106] to identify predicted transporters in the *A. baumannii* AB6870155 and *K. pneumoniae* KP6870155 genomes. The *K. pneumoniae* KP6870155 assembled genome was submitted to Kleborate[30] using the --kaptive parameter[107] to characterise its *K. pneumoniae* species complex (KpSC), ICE*Kp* virulence loci and plasmid virulence loci, K (capsule), O antigen (LPS) serotype and resistance genes. Needleman-Wuncsch pair-wise alignments were performed using ClustalW[64] and multiple sequence alignments (Fig. S14) were performed using Clustal Omega[108].

## Genomic identification of siderophores

To identify whether our strains contained commonly associated siderophores found in clinical isolates of *K. pneumoniae* (enterobactin, aerobactin, yersiniabactin) and *A. baumannii* (acinetobactin, baumannoferrin, fimsbactin), a fasta file containing coding sequences from these genes characterised in *A. baumannii* ATCC17978 (GenBank accession CP000521.1) and *K. pneumoniae* was searched against the genomes for AB6870155 and KP6870155 using BLASTn[109]. The Artemis[110] genome browser was used to visualise the genomic regions containing these siderophores to aid in constructing schematic representations of the genetic regions. Iron-related genes (Supplementary Data 11) were identified using FeGenie[111] version 1 (November 2020) with the supplied HMM database.

## Biolog phenotype microarrays

Biolog phenotype microarrays (PMs) were performed following the manufacturer's instructions[112] to identify compounds that could serve as sole carbon sources for AB6870155 and/or KP6870155 (PM1-2; 190 compounds). Separate 96-well plates containing MH or SLMM were also inoculated with mono- and co-cultures of *A. baumannii* AB6870155 and *K. pneumoniae* KP6870155, with and without tetrazolium-based dye to monitor cellular respiration and growth in these media (Fig. S4). Half the inoculum amount that was used for single strain inoculation was used for each strain when co-cultures were added to the MH and SLMM plates. After inoculation of the cells into the plates, they were incubated in an Omnilog incubator/reader (Biolog) for 48 h and read every 15 min for colour changes in each well associated with reduction of the tetrazolium-based dye (colourless) to formazan (violet) linked to cellular respiration. Data were analysed with the Omnilog-PM software, which generated a time-course curve for colour formation. An average height threshold of 101 arbitrary omnilog units (AOUs) was chosen to identify the carbon sources used by the strains. The growthcurver R package[113] was used to calculate the area under the logistic curve (auc_l). Scripts to generate plots were written in R (v.4.0.5) using the the ggplot2 package[114].

## Cross-feeding experiments using Millicell hanging cell culture inserts

Aliquots (50 µl) of the co-isolated bacteria from the overnight cultures, were inoculated into fresh MH medium with shaking to achieve the cell density of OD$_{600}$: 0.6, respectively. The KP6870155 was then inoculated into Millicell hanging insert (Merck) containing a sole carbon source. The insert only allowed the passage of small-secreted metabolites but not bacterial cells. AB6870155 was directly inoculated into 6-well plates containing the single carbon source that cannot be used by it and subsequently combined with inserts containing KP6870155. The assembled 6 well plates were incubated at 37 °C for 24 h with shaking. Fluorostar Omega spectrometer (BMG Labtech) was used to determine the growth of AB6870155. Each carbon source was assessed in triplicates.

## Successional co-feeding and HPLC

Successional co-feeding experiments were conducted in M9 media (Amresco, VWS Life Science, USA) supplemented with 2 mM MgSO$_4$, 0.1 mM CaCl$_2$ and 20 mM of either glycerol, lactose, mannitol, or sucrose, as the sole carbon source. An overnight culture of KP6870155 was grown in MH broth and subsequently washed three times in sterile M9 salts media. Washed cells were diluted to an OD600 0.01 using a Genesys 10 S UV-VIS spectrophotometer (ThermoScientific, USA) to read optical density and inoculated into M9 media supplemented with each carbon source; this was done in triplicate per carbon source for three timepoints including an uninoculated control. Tubes were tightly sealed and incubated at 37 °C statically creating a reduced oxygen environment. Absorbance readings were performed at the 8 h and 24 h post-inoculation timepoints using a CLARIOstarPlus plate reader (BMG Labtech, Germany). Also, at each post-inoculation timepoint, 1 ml was removed and centrifuged at 10,000 x g for 10 min at room temperature and the supernatant was filtered through a 0.22 µm filter and stored at −80 °C until later analysis. At 48 h post-inoculation, 100 µl of sample was removed for spectrophotometer reading, then the entire 15 ml culture was centrifuged at 10,000 x g for 10 min at room temperature. The supernatant was filtered through a 0.22 µm syringe filter and a 1 ml sample removed for analysis. The remaining ~13 ml of KP6870155-metabolised media was supplemented with M9 media, 2 mM MgSO$_4$ and 0.1 mM CaCl$_2$. A 10 ml aliquot was transferred to a fresh, sterile 50 ml tube to be used for AB6870155 growth, and the remaining ~2 ml kept as a control. AB6870155 was

grown in MH-broth overnight then 1 ml was removed and washed 3 times in sterile M9 media. Washed cells were used to inoculate the KP6870155-metabolised media with an inoculum diluted to OD600 0.01. Growth tubes, along with controls, were incubated at 37 °C statically, tightly sealed. Samples were taken at 8 h, 24 h, and 48 h post-inoculation as described above. The concentration of ethanol and lactate in samples was determined by high performance liquid chromatography (HPLC) on a Prominence UFLC system (Shimadzu, Japan), using an Aminex HPX-87H column (BioRad, USA) equipped with a refractive index detector. The column temperature was maintained at 60 °C with 0.01% (w/v) sulfuric acid as the mobile phase, injection volume of 10 µl and a flow rate of 0.6 ml/min. Data was analysed using the LabSolutions software (Version 5.54 SP3, Shimadzu Corporation). For successional cross-feeding of various *A. baumannii* strains (AB6870155, AB5075, BAL062) with spent media from various *K. pneumoniae* strains (KP6870155, SGH10, NTUH-K2044), all steps were performed as described above for collecting *K. pneumoniae* spent media and preparing overnight cultures of *A. baumannii*. At this point, 96-well microplate (Sarstedt) containing 100 µl of each *K. pneumoniae* spent media type was inoculated with washed *A. baumannii* cells at an $OD_{600} = 0.01$. Growth plates were incubated at 37 °C, 200 rpm, in a Clariostar plate reader (BMG Labtech) with $OD_{600}$ read every 10 min for 20 h. To prevent condensation on microplate lids and subsequent interference with readings, lids were treated with 0.05% Triton X-100 in 100% ethanol as described previously[115]. Growth plates had 3 technical replicates, with 3 biological replicates for each condition. Area under the curve (AUC) was calculated using the Growthcurver package (v0.3.1) in RStudio with default settings for whole plate growth curves[113]. Data processing and graphs were produced in RStudio using the tidyverse package (v1.3.1).

## RNA extraction and sequencing
AB6870155 and KP6870155 were grown as biofilms in synthetic lung mimicking medium (SLMM) separately and as a co-culture in in 6-well microtiter plates. After 24 h, the biofilms were washed with PBS buffer twice and biofilm associated cells collected. Total RNA was extracted using a RNeasy® Mini Kit (Qiagen®). RNA was quantified at 260 nm and its quality was assessed by agarose gel electrophoresis. Ribosomal RNA was depleted using the Ribo-Zero rRNA removal kit (Illumina®) according to the manufacturer's instructions prior to sequencing. mRNA library preparation and Illumina NextSeq-500 sequencing were conducted in the Ramaciotti Centre for Genomics (UNSW, Australia), and generated approximately 10 million paired end reads (75 bp read length) for each RNA sample.

## Transcriptomics analysis
RNA sequence reads were mapped to the respective hybrid assembled genomes using the STAR v 2.3.7a alignment tool[116] with parameters set to --sjdbGTFfeatureExon CDS and --genomeSAindexNbases 8 for the genomeGenerate command and --alignIntronMax 1 for the mapping command. Samtools v 1.9[117] was then used to sort and index the alignment files which were subsequently input into featureCounts[118] to generate counts tables. Read counts data was normalized and differential expression (DE) analysis conducted using the DESeq2 v 1.30.1 R package (65) with ashr shrinkage estimator[119]. Thus, read counts of samples were normalized for sequencing depth and distortion caused by highly differentially expressed genes. A negative binomial model was used to test the significance of differential expression between two conditions. A cutoff of FDR (False Discovery Rate) of less than 0.05 and a log2 fold change >1.0 was used to determine significantly differentially expressed genes. Differential gene expression across various cellular pathways was visualized in Biocyc[120].

## Biofilm quantitation assays
Biofilm formation assays were performed according to previously described method[121], with some alterations. 4–5 representative colonies from overnight streak plates on MH agar were resuspended in MH broth to equal CFUs ($5 \times 10^5$ cells/mL). These were inoculated into 96-well polystyrene microtitre plates (100 µl/well) as individual cultures and as a mixed co-cultures (at various ratios of the two strains) and plates were incubated for 18 h at 37 °C with and without 512 µg/mL cefotaxime. Quantification of the biofilm was carried out by measuring the amount of crystal violet stain retained in the wells (at an absorbance of 550 nm) with a Fluorostar Omega spectrometer (BMG Labtech, Offenburg, Germany). The results represent the average of three independent experiments, each with at least three technical replicates. Growth was quantified at $OD_{600nm}$ and the biofilm $OD_{550nm}$ to growth $OD_{600nm}$ ratio was calculated to determine the level of biofilm formed normalized to the total number of cells per sample.

## Confocal laser scanning microscopy (CLSM)
Biofilms were cultivated in flow-cell chambers as described previously[122] with some modifications. AB6870155 and KP6870155 strains were inoculated at similar CFUs individually or mixed, to flow-cell chambers (Biocentrum-DTU, individual channel dimensions: $1 \times 4 \times 40$ mm) covered with a glass coverslip, which served as a substratum for biofilm attachment. After inoculation of the bacterial suspension (500 µl) into the flow chamber, cells were incubated for 1 h at ambient temperature, after which MH broth was pumped into the flow-cell at a constant rate of 0.07 ml/min using an IPC-12 high precision peristaltic pump (Ismatec). Biofilms were prepared for CLSM after 3 days of growth by staining with Live/Dead BacLight bacterial viability probe (ThermoFisher) as per the manufacturer's protocol. A working concentration (500 µl) of the probe was injected gently into the flow-cell chambers to minimize disruption of the biofilms. The biofilms were stained for 15 min after which excess dye was removed by restarting the flow of growth medium and flushing for 30 min. Microscopic observation and image acquisition of biofilms was performed using the FV1000 CLSM (Olympus) equipped with argon and helium-neon lasers providing 488 and 543 nm excitation wavelengths, respectively. 510 nm to 530 nm interference and 610 nm long pass filters were used for emission. The captured biofilm images were further processed using Imaris software (Bitplane).

## Scanning electron microscopy (SEM)
For scanning electron microscopy (SEM), three colonies each of AB6870155 and KP6870155, making up three biological replicates each, from fresh streak plates on MH agar were inoculated into 10 ml of MH broth and grown overnight at 37 °C, with shaking at 200 rpm. Overnight cultures were pelleted by centrifugation at 5000 rpm and washed two times in SLMM. Washed pellets were resuspended in SLMM and normalized to $OD_{600}$ 1.0. Round coverslips were washed in acetone, dried completely and UV sterilised. The coverslips were then added to 6-well plates, one per well. 5 ml of SLMM media was added to each well, and subsequently each well was inoculated with 50 µl of normalized and washed replicate cultures. For the three wells designated for triplicate co-cultures, 25 µl of each strain AB6870155 and KP6870155 were inoculated into the wells. Plates were incubated for 21 h at 37 °C, with shaking at 100 rpm to allow for biofilm formation on the coverslips. Biofilms grown on coverslips were harvested at 21 h, washed gently three times with 0.01 M PBS. The samples were fixed with 3% glutaraldehyde for 1 h at ambient temperature followed by overnight incubation at 4 °C. Increasing gradients of ethanol washes were performed, 30% ethanol for 10 min, 50% ethanol 10 min, 70% ethanol 10 min, 80% ethanol 10 min, 90% ethanol 10 min., 100% ethanol 10 min, followed by critical drying on the Leica EM CPD300. Samples were gold sputter coated (Emitech K550) and visualised on the Phenom XL SEM (Thermo Scientific™). Images were analysed using

ImageJ software and the MicrobeJ plugin[47] for quantitative measurements of cell length and statistical testing. Beeswarm plots representing this data were generated with beeswarm (v 0.4.0) and Rserve (1.8–10) R packages.

## In-vivo *Galleria mellonella* trials

The *G. mellonella* injection protocol was performed as previously described[123] with some modifications. Three technical replicate assays comprising of 5 larvae (200–250 mg) each were performed using $1 \times 10^5$ cells of either *A. baumannii* AB6870155 and *K. pneumoniae* KP6870155 for single infections. For coinfections a mixture of $1 \times 10^5$ of each AB6870155 and KP6870155 was made prior to injection. Following injection, the larvae were incubated at 37 °C and examined each day for up to 10 days post-injection and scored for survival according to the *G. mellonella* Health Index Scoring System[124]. In a separate batch trial, three to four technical replicates comprising of 5 larvae were performed using $1 \times 10^7$ cells of either strain. For coinfections a mixture of $5 \times 10^6$ of each AB6870155 and KP6870155 was made prior to injection. The larvae for all trials were of the same age (fifth instar larval stage). Following injection, the larvae were incubated at 37 °C and examined each day for up to 10 days post-injection and scored for survival. Survival analysis was performed using the Survminer (v 0.4.9) R package.

## Statistics and reproducibility

The statistics was done using R statistical software and Microsoft Excel (version 16.67). Experiments were performed with three independent culture replicates and similar results were obtained for all.

## Reporting summary

Further information on research design is available in the Nature Portfolio Reporting Summary linked to this article.

## Data availability

The sequence reads were submitted to GenBank under project number PRJNA263680. *A. baumannii* AB6870155 genome BioSample accession SAMN03105183, *K. pneumoniae* KP6870155 genome BioSample accession SAMN23708555. RNA sequence reads were deposited in the NCBI Sequence Read Archive (SRA) under accession numbers SRR20379129, SRR20379130, SRR20379131, SRR20379132, SRR20379133, SRR20379134, SRR20379135, SRR20379136, SRR20379137. Source data are provided with this paper. Supplementary Data files 1–12 are accessible via GitHub at https://github.com/amycainlab/coinfection_project. Source data are provided with this paper.

## Code availability

Scripts and bioinformatics pipeline can be accessed via GitHub at https://github.com/amycainlab/coinfection_project and are available via Zenodo (https://doi.org/10.5281/zenodo.7430526).

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

## Acknowledgements

We would like to thank Professor Melissa Brown from Flinders University for providing the strains. We would like to thank Sue Lindsay from Macquarie University Microscopy Unit for her assistance with SEM. This work was partially supported by the National Health and Medical Research Council (NHMRC) Australia through Project Grant 1159752 to A.K.C., NHMRC Grants 1124917, 1120298, 1060895 to I.T.P. and a University of Newcastle CESE Excellence grant to K.A.H., A.K.C., L.S. and C.J.D. A.K.C. is supported by an Australian Research Council DECRA fellowship (DE180100929). K.A.H. is supported by an Australian Research Council Future fellowship (FT180100123).

## Author contributions

All authors conceived of various aspects in the study. L.S. and H.L. performed the cross-protection assay. Q.L. performed the RNA-seq, Biolog PM and cross-feeding experiments. L.S. performed genome sequencing and RNA-seq and genome analysis. C.D. performed the successional co-feeding and HPLC experiment and C.D. and L.S. analysed the data. C.D. performed the qPCR experiments. A.P. and L.S. conducted the SEM experiment and analysed the data. A.P. and Q.L. performed the confocal microscopy and analysis. H.L. performed the biofilm tests and H.L. and L.S. analysed the data. H.D. and R.M. performed the *G. mellonella* in vivo infection experiments and H.D., L.S., R.M. and A.K.C. analysed the data. F.L.S. contributed to bioinformatic analysis using Kleborate and FeGenie analysis tools. I.T.P., K.A.H. and A.K.C. contributed funding. L.S., A.K.C., I.T.P. and K.A.H. wrote the manuscript. All authors reviewed and edited the manuscript.

## Competing interests

The authors declare no competing interests.

## Ethics approval

*The Klebsiella pneumonia* KP6870155 and *Acinetobacter baumannii* AB6870155 isolates were isolated from leftover deidentified samples collected during routine hospital-associated diagnostic testing[1]. As a result, neither informed consent nor approval from an ethics committee was required.
