## [Peer Review File · Nature Communications]

Cross-protection and cross-feeding between *Klebsiella pneumoniae* and *Acinetobacter baumannii* promotes their co-existenceREVIEWER COMMENTS

Reviewer #1 (Remarks to the Author):

The purpose of this study was to investigate synergistic interactions between *Klebsiella pneumoniae* and *Acinetobacter baumannii* isolated from a human lung infection. The authors perform a range of assays, from genetic to phenotypic, to investigate the nature of this interaction. They find that these pathogens are able to cross-feed and cross-protect from antibiotic exposure. This work provides novel and critical insights into the mechanisms of polymicrobial interactions in infection contexts.

This paper has several strong points that I would like to highlight. The first is that the two strains used in the paper are clinical isolates from a single patient; this supports the idea that the findings in this paper are relevant to clinical infection contexts. Second, the authors' integration of RNAseq data into motivations for, and explanations of, phenotypic experiments is excellent. Third, the authors use a variety of both planktonic and biofilm assays to phenotype their co-cultures, which produces a comprehensive view of the interactions between these taxa. Overall, this study is very well done and makes some interesting conclusions about how taxa might interact in infection contexts. My other comments on the paper are below.

Major comments:

- Line 468-469 Are the authors able to distinguish between cross-protection against antibiotics due to biofilm formation vs due to cephalosporinase secretion? One way to investigate this would be to repeat the cross-feeding experiment described in lines 262-263, but add antibiotics to the culture. A cephalosporinase might be able to diffuse across the 0.2um filter and provide cross-protection (if it is an extracellular enzyme) and/or degrade the antibiotic in the medium such that KP6870155 experiences less antibiotic exposure. If no cross-protection is observed here, that would suggest that the cross-protection is mostly due to biofilm formation (or some other contact-dependent phenomenon).
- There are several points in the paper where the authors indicate that further study is required on a particular phenomenon they observed (e.g. lines 147-148 on the role of hypothetical genes in coinfection adaptation, lines 574-575 on the TonB-dependent siderophore production in co-culture growth *A. baumannii*). It would be helpful to have a paragraph in the discussion section elaboration on these points. Additionally, a brief discussion on the limitations of this work should be included.
- Given that these isolates were obtained from patient samples, could the authors discuss the implications of their work in management of polymicrobial infections in human patients?

Minor comments:

- Line 72-75: this sentence is somewhat confusing; recommend rewording or breaking into two sentences.
- It is not always clear in the methods section which assays are performed anaerobically vs aerobically. For example, in line 824-825, does "tubes were tightly sealed" mean the experiment was performed under anerobic conditions? What about the overnight cultures in this experiment?
- Figure S3: given that the terms "persistent", "shell", and "cloud" genes are specific to PPanGGOLiN, it would be helpful to have these terms defined in the figure legend for a reader who might not be familiar with these terms. These terms could also be explained in the results section (around line 133-134).
- Line 162-164: Table S10 is referenced immediately after table S6; recommend re-ordering these tables so they are referenced in numerical order in the text.
- Line 276: why was lactate of interest in this experiment? The motivation for looking at ethanol is explained in the previous paragraph, but lactate is not. Lactate is mentioned around line 239-240, but it would be helpful to remind the reader why lactate is being looked at here- especially since no change in lactate transporter expression was observed.
- Line 276: were these experiments performed in planktonic culture or during biofilm growth (for both KP and AB)?
- Line 285-286: I recommend spelling out explicitly here that any reduction in ethanol or lactate observed in figures 4b,d,f,h is a result of AB consuming spent medium components from KP metabolism.
- Line 420: was the MIC testing performed in planktonic or biofilm cultures? The MICs from one

are not necessarily directly comparable to the other, so it is relevant to clarify this point.

- Line 499-500: the decreased expression of T6SS genes doesn't necessarily indicate a lack of competition. Not all competitive interactions between bacteria (even those with T6SS systems in their genomes/on plasmids) involve T6SS machinery.
- Line 586: why was *Galleria mellonella* chosen as an infection model?
- Line 587: the authors state that 10^5 cells of KP6870155 and AB6870155 were injected individually and mixed in equal proportions prior to the injection. Does this mean the total number of cells injected was kept constant (i.e. 50,000 KP cells and 50,000 AB cells in the co-infection) or the total number of each species was kept constant (i.e. 10^5 KP cells and 10^5 AB cells in the co-infection)?

Reviewer #2 (Remarks to the Author):

This is a superbly written manuscript that begins with a description of the genetic composition of an *A. baumannii* and a *K. pneumoniae* strain that were co-isolated from a single lung infection. Most of the study focuses on transcriptional analyses of each organism in isolation or in combination during planktonic and biofilm growth states. The data are exquisitely described in terms of metabolic, virulence factor, quorum sensing, and antibiotic resistance gene expression, which led to the novel finding that *K. pneumoniae* metabolism is likely to generate lactate and/or ethanol that can be used by- and promote- growth of *A. baumannii* during co-infection. Conversely, *A. baumannii* provides cefotaxime cross protection toward *K. pneumoniae* (presumably via *A. baumannii* beta-lactamase production), which is interesting and supported by literature. The authors conclude that the collective transcriptome changes in co-culture conditions indicate intimate relationships exist between the organisms in a manner that may increase their disease capability and use a wax worm model of infection to indeed show that co-inoculated worms are slightly less tolerant of the organisms than when inoculated with each bacterial species in isolation.

Overall, the work is very nicely organized and the results are objectively interpreted and described. At issue, the study is descriptive in nature- in iterative manner a set of transcriptome data is presented and a phenotypic study approaches verifying the RNA data, which is nice. But the work lacks verification (i.e. do ethanol and/or lactate actually promote *A. baumannii* growth?), it also lacks mechanism of action studies that account for phenotypes seen (i.e. does *A. baumannii* release B-lactamases) and is largely confirmatory of other's work (i.e. genes of a particular network are co-regulated). While this reader likes the work very much I do find it well below the impact needed for publication in a journal with diverse readership & suggest it would be better suited for a more specialized journal.

Reviewer #3 (Remarks to the Author):

The authors present a very intriguing study suggesting that *Acinetobacter* and *Klebsiella* can impact each other's behavior when in mixed infections. This could have a significant impact on infections. However, there are numerous concerns that require clarification and some experiments.

Major comments

1) The data suggesting there is a significant alteration of bacterial growth in coculture are not clear. In Figures 2b and 2c, it would be helpful to separately graph the levels of *Klebsiella* and *Acinetobacter* within the coculture, as well as the total number of bacteria combined (as is currently done). Then it would be possible to directly compare *Klebsiella* monoculture with *Klebsiella* within the co-culture, and the same for *Acinetobacter*. At present, it is not very clear that there is a significant growth advantage in the coculture. Could the curves also be interpreted to show that *Klebsiella* inhibits *Acinetobacter* growth, rather than *Acinetobacter* promoting *Klebsiella* growth?

In addition, the differences between Figures 2d and 2e look minimal. For these panels, it would be helpful to have a time zero included so we can understand how the population shifted compared to the starting point.

2) The data in this study are based on one isolate of *Acinetobacter* and one isolate of *Klebsiella*. It is critical that the authors investigate whether other clinical isolates of each bacterium behave similarly. It may not be possible to easily acquire isolates from the same patient as was the origin of these current isolates. However, that is not a requirement. It is doubtful that this is a phenotype only exhibited by isolates that have co-infected the same patient. If that is somehow true it would be an interesting and important result. But I imagine if this phenotype is widespread it should occur with other isolates, even if they originated from single infections.

3) The experiments examining metabolic profiles are interesting. If possible, the experiments showing "cross-feeding" would be enhanced by using defined metabolic pathway mutants that cannot utilize certain carbon sources.

4) In Figure 5f there is maybe a 2-fold increase in average biofilm size with co-culture. Is it certain that this difference is biologically relevant?

5) In Figure 6b the combination of bacteria is more lethal, but twice as many cfu are being inoculated. As a control it is important to test inoculation of 2×10^5 of each single bacterium, to separate the inoculum amount from the single or combined makeup of the inoculum.

6) Are *Klebsiella* exposed to *Acinetobacter* supernatants more antibiotic resistant? Or is the resistance due to mixed biofilm structure? The mixed culture resistance was to the level of resistance of the *Acinetobacter*. Is this a feature of the metabolic interaction of the strains, or just the fact that this one *Acinetobacter* is more resistant? If a susceptible *Acinetobacter* strain is used, would there be any protection?

7) Figure 6a – What is the proportion of starting *Klebsiella* cells that survived? The picture is nice but it does not give any indication of the magnitude of the killing experienced by *Klebsiella*. Thus, we cannot have any measure of the protection conferred upon *Klebsiella* by *Acinetobacter*. What happens at later timepoints? Are all the *Klebsiella* killed? Whether the *Klebsiella* are truly protected is unclear. Again Figure 6a would need a time zero as well to judge this.

8) The methods state that for co-culture experiments the different bacteria were mixed at "defined ratios". In Table 2 this ratio is not mentioned. In the legend for Figure 6a this ratio is not mentioned. It is very difficult to completely understand what is being done in some of these experiments.

Minor comments

1) The way these sentences are written it sounds like *Klebsiella* is not CRE, but it is. "Dual-infections by *Acinetobacter baumannii* or *Pseudomonas aeruginosa* in patients with carbapenem-resistant Enterobacteriaceae (CRE) have been shown to have increased antibiotic resistance levels and mortality rates compared to single infections. Similarly, increased mortality has also been observed in critically ill patients co-infected with *Klebsiella pneumoniae*, *P. aeruginosa* and/or *A. baumannii*."

2) The legend for Figure 5b does not specify which parameter (growth or biofilm mass) is indicated by the bars or the black points.

Reviewer 1

Major comments:

1) Line 468-469 Are the authors able to distinguish between cross-protection against antibiotics due to biofilm formation vs due to cephalosporinase secretion? One way to investigate this would be to repeat the cross-feeding experiment described in lines 262-263, but add antibiotics to the culture. A cephalosporinase might be able to diffuse across the 0.2um filter and provide cross-protection (if it is an extracellular enzyme) and/or degrade the antibiotic in the medium such that KP6870155 experiences less antibiotic exposure. If no cross-protection is observed here, that would suggest that the crossprotection is mostly due to biofilm formation (or some other contact-dependent phenomenon).

- We thank the reviewer for recommending this experiment to help distinguish the mode of action for cross-protection. We performed the Millicell hanging insert experiment with MH medium + cefotaxime at various concentrations (16-512 ug/mL). *A. baumannii* AB6870155 was inoculated in hanging inserts and *K. pneumoniae* KP6870155 was inoculated into wells ensuring physical separation of the two species but with the ability to exchange extracellular enzymes and antibiotics in media. We measured the absorbance of the cultures after 21 hr of growth and found that the OD600 fell to zero at 32 ug/mL in the KP only wells (that did not contain an insert with AB) while the OD600 significantly increased for wells with KP that also contained an AB inoculated insert (see figure S10b below).
- This result indicates some cross-protection is conferred by cephalosporinase.
- We have incorporated this result into the paper in the Methods, Results and Supplementary Figure S10b (see below – lines revised are shown in screenshots)
- Note: Figure S10a was added in response to reviewer 3 comment #8.

Figure S10 – Cross-protection in physically interacting and separated co-cultures of *A. baumannii* and *K. pneumoniae*. **a**, Survival measured by colony counts (CFU/mL) of mono-cultures, AB6870155 (AB) and KP6870155 (KP), and co-cultures of AB6870155 + KP6870155 (AB+KP) at 70:30 ratio after initial (0 hr) and 21 hr of exposure to 512 µg/mL cefotaxime. Red colony counts corresponding to KP6870155 on MacConkey agar in the co-culture plated cells are represented by KP (AB+KP) and white colony counts corresponding to AB6870155 in the co-culture plated cells are represented by AB (AB+KP). **b**, Growth in various concentrations of cefotaxime measured by optical density (OD 600nm) of mono-cultures of KP6870155 (KP only) and physically separated (via Millicell hanging insert) co-cultures of KP6870155 and AB6870155 (KP + AB insert).

- Revised results section:

510 µg/ml to 8 µg/ml (Fig. 6a). To further elucidate the mode of cross-protection, we performed a
 511 Millicell hanging insert experiment where KP6870155 was physically separated from
 512 AB6870155 but could share the same media and any secreted molecules therein. Growth of
 513 KP6870155 was significantly higher when sharing media with AB6870155 than when grown
 514 alone (Fig. S10b). These results indicate there is cross-protection of KP6870155 by
 515 AB6870155, likely through cephalosporinase activity.

- Revised methods section:

794 Cross-protection experiments were also performed using Millicell hanging inserts (Millipore)
 795 where *A. baumannii* AB6870155 was inoculated into inserts that were placed into wells of a
 796 24-well plate inoculated with *K. pneumoniae* KP6870155 at 5 x 10⁵ CFU/ml. Various
 797 concentrations of cefotaxime (16-512 µg/ml) were tested and after 21 hr incubation at 37°C,
 798 cell growth was measured at OD₆₀₀ with a spectrophotometer.

2) There are several points in the paper where the authors indicate that further study is required on a particular phenomenon they observed (e.g. lines 147-148 on the role of hypothetical genes in coinfection adaptation, lines 574-575 on the TonB-dependent siderophore production in co-culture growth *A. baumannii*). It would be helpful to have a paragraph in the discussion section elaboration on these points. Additionally, a brief discussion on the limitations of this work should be included.

We appreciate the reviewer's comment. Keeping in mind word count limitations, we have addressed this comment succinctly as follows in the Results section:

156 were hypothetical genes for KP6870155. The roles of these genes may have not yet been
157 identified in coinfection adaptation which is a limitation of this work and warrants further
158 study.

637 despite the presence of *K. pneumoniae* as an additional competitor for iron. Besides iron
638 acquisition genes found in the major siderophore operons, bacteria also contain siderophore
639 receptor genes distributed more randomly in the genome that can recognize exogenously
640 produced siderophores⁷⁷. In line with this, a TonB-dependent siderophore receptor gene,
641 NH10_03345, had over 2-fold increased expression in co-culture grown *A. baumannii* and its
642 ability to uptake xenosiderophores warrants further investigation as TonB-dependent receptors
643 are known to transport xenosiderophores in many Gram-negatives including *A.baumannii*⁷⁸.

3) Given that these isolates were obtained from patient samples, could the authors discuss the implications of their work in management of polymicrobial infections in human patients?

- We thank the reviewer for this comment and have added the following to the Discussion section:

759 pathogens co-isolated from polymicrobial infections in clinical settings. This work
760 demonstrates the need to identify polymicrobial infections and test coinfecting pathogens for
761 cross-protection prior to antibiotic administration. This will ensure the most effective regimen
762 can be assigned that has minimal likelihood for recalcitrance to therapy caused by cross-
763 protection between coinfecting pathogens.

Minor comments:

4) Line 72-75: this sentence is somewhat confusing; recommend rewording or breaking into two sentences.

- We have split this into two sentences to increase its clarity (see below):

74 with transporters that can efflux these cationic hydrocarbons out of the cell ¹⁰. In terms of their
75 interactions with host cells, *K. pneumoniae* produces two well characterised cell surface
76 polysaccharides, lipopolysaccharide O antigen and polysaccharide capsule (K) and less well
77 elucidated enterobacterial common antigen, which allow it to evade host immune attacks ^{20,21}.
78 *A. baumannii* forms lipooligosaccharide and capsule but lacks lipopolysaccharide, due to the
79 absence of an O-antigen ligase ²². The *A. baumannii* capsular polysaccharide (encoded by the

5) It is not always clear in the methods section which assays are performed anaerobically vs aerobically. For example, in line 824-825, does “tubes were tightly sealed” mean the experiment was performed under anerobic conditions? What about the overnight cultures in this experiment?

- We thank the reviewer for this comment and have added clarification to the Methods sections (see below).

770 this study, cells were collected in mid-log phase after being sub-cultured from overnight cell
771 cultures grown aerobically at 37°C with shaking at 200 rpm. Mueller-Hinton (MH) medium, a

924 per carbon source for three timepoints including an uninoculated control. Tubes were tightly
925 sealed and incubated at 37°C statically creating a reduced oxygen environment. Absorbance

6) Figure S3: given that the terms “persistent”, “shell”, and “cloud” genes are specific to PPanGGOLiN, it would be helpful to have these terms defined in the figure legend for a reader who might not be familiar with these terms. These terms could also be explained in the results section (around line 133-134).

- Unlike other methods that use an arbitrary threshold to define persistent, shell and cloud genomes, PPanGGOLiN relies on a statistical model for the partitioning that combines information on the presence/absence of gene families and their genetic contiguity in the pangenome graph. This model was shown to be more robust in estimating the persistent (soft-core) genome than the classical threshold of 95% of presence. Therefore there is not an exact definition of the threshold for cloud, shell and persistent (ie. persistent is present in >90% genomes, cloud present in <15%, shell in 15-90%) as in other tools.
- However, to clarify the broader terms used for the reader, we have added other common names given for persistent (aka soft-core), shell (aka core/shell) and cloud (aka accessory/dispensable) genes, that are used by other pangenome analysis tools like Roary and PanGP in the Figure S3 legend and in text to give readers a more familiar name for these terms (see below for revised text).

135 sequences available in RefSeq, downloaded on September 30, 2020 (Table S4). Of the total
 136 number of genes in AB6870155, 12.4% were cloud (accessory/dispensable) genes while the

152 PPanGGOLiN, which defines a RGP as a set of consecutive genes that are part of the shell
 153 (core) or cloud genomes. *A. baumannii* AB6870155 contained 33 RGPs, one of which was

Figure S3 – Pangenomes of AB6870155 and KP6870155. Distribution of persistent (soft-core), shell (core/shell) and cloud (accessory/dispensable) genes for **a**, AB6870155 compared to **b**, pangenome containing 173 strains of *A. baumannii* and **c**, KP6870155 compared to **d**, pangenome containing 573 strains of *K. pneumoniae*. Pangenomes computed by PPanGGOLiN (20).

7) Line 162-164: Table S10 is referenced immediately after table S6; recommend reordering these tables so they are referenced in numerical order in the text.

- We thank the reviewer for this observation and have re-numbered the tables, so they are referenced in numerical order in text.

8) Line 276: why was lactate of interest in this experiment? The motivation for looking at ethanol is explained in the previous paragraph, but lactate is not. Lactate is mentioned around line 239-240, but it would be helpful to remind the reader why lactate is being looked at here- especially since no change in lactate transporter expression was observed.

- Lactate was specifically mentioned because it is another possible by-product of glucose fermentation in addition to ethanol. Also, the SLMM

medium that the co-cultures were grown in for the experiments contained lactate and glucose as carbon sources. We have made clarified this in the text as follows:

259 its competing fermentation end-product, lactate (which is also provided as carbon source in
260 SLMM), could be produced in the co-culture biofilms. Concordantly, *A. baumannii*

9) Line 276: were these experiments performed in planktonic culture or during biofilm growth (for both KP and AB)?

- These experiments were performed in biofilm conditions, and we have added this in the text as follows:

304 during metabolism of four carbon sources utilised by KP6870155 but not AB6870155:
305 glycerol, lactose, mannitol and sucrose under biofilm conditions. We confirmed that

10) Line 285-286: I recommend spelling out explicitly here that any reduction in ethanol or lactate observed in figures 4b,d,f,h is a result of AB consuming spent medium components from KP metabolism.

- We thank the reviewer for this suggestion and have incorporated the interpretations of these Figures into the text as follows:

312 filtered spent media containing KP6870155 metabolites. Hence, the reduction in ethanol or
313 lactate observed (Fig. 4b,d,f,h) is a result of AB6870155 consumption of these KP6870155
314 metabolic by-products. The consumption of these by AB6870155 is likely given the ability of
315 *A. baumannii* to grow in L-lactic acid (Table S9) and the previously reported ethanol utilization
316 by *A. baumannii*³⁵. A reduction in ethanol was observed during growth of AB6870155 in

11) Line 420: was the MIC testing performed in planktonic or biofilm cultures? The MICs from one are not necessarily directly comparable to the other, so it is relevant to clarify this point.

- These experiments were performed in biofilm conditions, and we have added this information in the text as follows:

457 we tested whether the co-infection isolates exhibited any cross-protection against antibiotics
458 during biofilm conditions. We first established the MICs of both mono and co-cultures (1:1

12) Line 499-500: the decreased expression of T6SS genes doesn't necessarily indicate a lack of competition. Not all competitive interactions between bacteria (even those with T6SS systems in their genomes/on plasmids) involve T6SS machinery.

- We thank the reviewer for this comment and have clarified this in text (see below) to indicate the competition is not specifically T6SS-*based* leaving the possibility for other forms of competition open.

576 The decreased expression levels of most T6SS genes in co-grown biofilms with KP6870155

577 indicate a lack of T6SS-based competition between these two bacteria.

13) Line 586: why was *Galleria mellonella* chosen as an infection model?

- *G. mellonella* has previously been used as an infection model for understanding various microbial co-infections as we have reviewed recently (doi: 10.1093/femspd/ftab006). We have added this clarification to the text as follows:

660 model. *G. mellonella* was chosen as an infection model given its suitability for studying

661 bacterial co-infections⁸¹. 10⁵ cells of *K. pneumoniae* KP6870155 and *A. baumannii*

14) Line 587: the authors state that 10⁵ cells of KP6870155 and AB6870155 were injected individually and mixed in equal proportions prior to the injection. Does this mean the total number of cells injected was kept constant (i.e. 50,000 KP cells and 50,000 AB cells in the co-infection) or the total number of each species was kept constant (i.e. 10⁵ KP cells and 10⁵ AB cells in the co-infection)?

- The total number of each species was kept constant (10⁵ KP + 10⁵ AB), giving a final inoculum size of 2x10⁵ for the injected cocultures. This has been clarified in the text as follows:

662 AB6870155 were injected individually and mixed in equal proportions (with the total number

663 of each species kept constant at 10⁵ cells each) just prior to injection and larvae survival over

Reviewer #2 (Remarks to the Author):

This is a superbly written manuscript that begins with a description of the genetic composition an *A. baumannii* and a *K. pneumoniae* strain that were co-isolated from a single lung infection. Most of the study focuses on transcriptional analyses of each organism in isolation or in combination during planktonic and biofilm growth states. The data are exquisitely described in terms of metabolic, virulence factor, quorum sensing, and antibiotic resistance gene expression, which led to the novel finding that *K. pneumoniae* metabolism is likely to generate lactate and/or ethanol that can be used by- and promote growth of *A. baumannii* during co-infection. Conversely, *A. baumannii* provides cefotaxime cross protection toward *K. pneumoniae* (presumably via *A.*

baumannii beta-lactamase production), which is interesting and supported by literature. The authors conclude that the collective transcriptome changes in co-culture conditions indicate intimate relationships exist between the organisms in a manner that may increase their disease capability and use a wax worm model of infection to indeed show that co-inoculated worms are slightly less tolerant of the organisms than when inoculated with each bacterial species in isolation. Overall, the work is very nicely organized and the results are objectively interpreted and described. At issue, the study is descriptive in nature- in iterative manner a set of transcriptome data a presented and a phenotypic study approaches verifying the RNA data, which is nice. But the work lacks verification (i.e. do ethanol and or lactate actually promote *A. baumannii* growth?), it also lacks mechanism of action studies that account for phenotypes seen (i.e. does *A. baumannii* release B-lactamases) and is largely confirmatory of other's work (i.e. genes of a particular network are co-regulated). While this reader likes the work very much I do find it well below the impact needed for publication in a journal with diverse readership & suggest it would be better suited for a more specialized journal.

We would like to thank the reviewer for their supportive feedback and also for highlighting concerns regarding verification of mode of action and ability of *A. baumannii* to grow with ethanol and lactate as sole carbon sources. We have addressed these concerns as follows:

Mode of action addressed

- In response other reviewer's comments, we performed the Millicell hanging insert experiment and found that cephalosporinases have a role in the mode of action for cross-protection. The experiment was performed with MH medium + cefotaxime at various concentrations (16-512 ug/mL). *A. baumannii* AB6870155 was inoculated in hanging inserts and *K. pneumoniae* KP6870155 was inoculated into wells ensuring physical separation of the two species but with the ability to exchange extracellular enzymes and antibiotics in media. We measured the absorbance of the cultures after 21 hr of growth and found that the OD600 fell to zero at 32 ug/mL in the KP only wells (that did not contain an insert with AB) while the OD600 significantly increased for wells with KP that also contained an AB inoculated insert (see figure S10b below).
- This result indicates some cross-protection is conferred by cephalosporinase.
- We have incorporated this result into the paper in the Methods, Results and Supplementary Figure S10b (see below).
- Note: Figure S10a was added in response to reviewer 3 comment #8.

Figure S10 – Cross-protection in physically interacting and separated co-cultures of *A. baumannii* and *K. pneumoniae*. **a**, Survival measured by colony counts (CFU/mL) of mono-cultures, AB6870155 (AB) and KP6870155 (KP), and co-cultures of AB6870155 + KP6870155 (AB+KP) at 70:30 ratio after initial (0 hr) and 21 hr of exposure to 512 µg/mL cefotaxime. Red colony counts corresponding to KP6870155 on MacConkey agar in the co-culture plated cells are represented by KP (AB+KP) and white colony counts corresponding to AB6870155 in the co-culture plated cells are represented by AB (AB+KP). **b**, Growth in various concentrations of cefotaxime measured by optical density (OD 600nm) of mono-cultures of KP6870155 (KP only) and physically separated (via Millicell hanging insert) co-cultures of KP6870155 and AB6870155 (KP + AB insert).

- Revised results section:

510 µg/ml to 8 µg/ml (Fig. 6a). To further elucidate the mode of cross-protection, we performed a
 511 Millicell hanging insert experiment where KP6870155 was physically separated from
 512 AB6870155 but could share the same media and any secreted molecules therein. Growth of
 513 KP6870155 was significantly higher when sharing media with AB6870155 than when grown
 514 alone (Fig. S10b). These results indicate there is cross-protection of KP6870155 by
 515 AB6870155, likely through cephalosporinase activity.

- Revised methods section:

794 Cross-protection experiments were also performed using Millicell hanging inserts (Millipore)
 795 where *A. baumannii* AB6870155 was inoculated into inserts that were placed into wells of a
 796 24-well plate inoculated with *K. pneumoniae* KP6870155 at 5 x 10⁵ CFU/ml. Various
 797 concentrations of cefotaxime (16-512 µg/ml) were tested and after 21 hr incubation at 37°C,
 798 cell growth was measured at OD₆₀₀ with a spectrophotometer.

A. baumannii growth in ethanol/lactate addressed

A. baumannii is capable of utilising ethanol as a carbon source (Camarena, L., *et al*, 2010, PLoS Pathogens). Additionally, we found that *A. baumannii* was able to grow on lactate (AUC 11420) (Table S9). We have clarified this in the results as follows:

312 filtered spent media containing KP6870155 metabolites. Hence, the reduction in ethanol or
313 lactate observed (Fig. 4b,d,f,h) is a result of AB6870155 consumption of these KP6870155
314 metabolic by-products. The consumption of these by AB6870155 is likely given the ability of
315 *A. baumannii* to grow in L-lactic acid (Table S9) and the previously reported ethanol utilization
316 by *A. baumannii*³⁵. A reduction in ethanol was observed during growth of AB6870155 in

Reviewer #3 (Remarks to the Author):

The authors present a very intriguing study suggesting that *Acintobacter* and *Klebsiella* can impact each others behavior when in mixed infections. This could have a significant impact on infections. However, there are numerous concerns that require clarification and some experiments.

Major comments

1) The data suggesting there is a significant alteration of bacterial growth in coculture are not clear. In Figures 2b and 2c, it would be helpful to separately graph the levels of *Klebsiella* and *Acintobacter* within the coculture, as well as the total number of bacteria combined (as is currently done). Then it would be possible to directly compare *Klebsiella* monoculture with *Klebsiella* within the co-culture, and the same for *Acintobacter*. At present, it is not very clear that there is a significant growth advantage in the coculture.

- We thank the reviewer for their suggestion. Although we had tried separately graphing growth of each species in co-culture by performing colony counts throughout growth on differential agar, we had encountered technical issues that made it difficult to confidently attribute cell numbers to each species. Cultures were visibly aggregating, which lead to abnormal serial dilution patterns between replicates yielding poor reproducibility using this method (see example figures below). Hence, we have instead performed qPCR to quantify each species abundance during growth in co-culture. We used controls to ensure the amount of DNA being extracted from both species was consistent as described in our methods.

Could the curves also be interpreted to show that Klebsiella inhibits Acinetobacter growth, rather than Acinetobacter promoting Klebsiella growth?

- It is difficult to draw either conclusion. Instead, based on the RNA-seq data, it appears Acinetobacter has increased respiratory activity when co-cultured with Klebsiella in SLMM which was also evident by the increase in respiratory activity of co-cultures in SLMM media. We have clarified the link between these observations as follows:

218 phase (24 h) when grown in MH. *A. baumannii* AB6870155 mono-cultures had a less
 219 pronounced growth advantage (AUC 6819) over *K. pneumoniae* KP6870155 mono-cultures
 220 (AUC 4856) when grown in SLMM (Fig. 2b) and the respiratory activity of co-cultures (AUC
 221 17439) was higher than that of AB6870155 mono-cultures in SLMM (AUC 17033) (Fig. 2b)
 222 which corresponded with the increased expression of respiratory pathways in *A. baumannii* co-
 223 cultured with *K. pneumoniae*. It is likely that most of the respiratory activity in co-cultures can

In addition, the differences between Figures 2d and 2e look minimal. For these panels, it would be helpful to have a time zero included so we can understand how the population shifted compared to the starting point.

- Time zero was added to figures 2d and 2e (see below)

2) The data in this study are based on one isolate of *Acintobacter* and one isolate of *Klebsiella*. It is critical that the authors investigate whether other clinical isolates of each bacterium behave similarly. It may not be possible to easily acquire isolates from the same patient as was the origin of these current isolates. However, that is not a requirement. It is doubtful that this is a phenotype only exhibited by isolates that have co-infected the same patient. If that is somehow true it would be an interesting and important result. But I imagine if this phenotype is widespread it should occur with other isolates, even if they originated from single infections.

- We thank the reviewer for their recommendation of these confirmational experiments. We have accordingly performed successional co-feeding experiments with AB6870155 and other *A. baumannii* strains (BAL062 and AB5075 – both human origin) in filtered spent media from various *K. pneumoniae* strains including ATTC 43816, SGH10 and NTUHK2044 (all of which are human origin). The *K.*

pneumoniae strains were initially grown with various defined carbon sources (Gly = glycerol, Lac = lactose, Mann = mannitol, Suc = sucrose) and then the spent medium was filter sterilised and fed to the *A. baumannii* strains. The results indicate that this cross-feeding phenotype is indeed widespread across most clinical strains tested albeit not ATCC 43816. These results were incorporated into a Supplementary Figure and added to the Methods and Results sections (see below).

Figure S8 – Successional co-feeding of *A. baumannii* strains with spent media from various *K. pneumoniae* strains. Growth is represented by area under curve (AUC) measurements taken from growth curves of *A. baumannii* strains (BAL062, AB5075 and AB6870155) grown in filtered spent media from *K. pneumoniae* strains (KP6870155, SGH10, NTUH-K2044 and ATCC 43816) fed with defined carbon sources; Gly = glycerol, Lac = lactose, Mann = mannitol, Suc = sucrose.

- Revised results section:

330 Furthermore, this cross-feeding phenotype appears to also be widespread amongst other *A.*
 331 *baumannii* (AB5075, BAL062) and *K. pneumoniae* (ATCC 43816, NTUHK2044, SGH10)
 332 isolates tested and is not only seen for isolates specifically from co-infections (Fig. S8).

- Revised methods section:

945 using the LabSolutions software (Version 5.54 SP3, Shimadzu Corporation). For successional
946 cross-feeding of various *A. baumannii* strains (AB6870155, AB5075, BAL062) with spent
947 media from various *K. pneumoniae* strains (KP6870155, SGH10, NTUH-K2044), all steps
948 were performed as described above for collecting *K. pneumoniae* spent media and preparing
949 overnight cultures of *A. baumannii*. At this point, 96-well microplate (Sarstedt) containing 100
950 µl of each *K. pneumoniae* spent media type was inoculated with washed *A. baumannii* cells at
951 an OD₆₀₀ = 0.01. Growth plates were incubated at 37°C, 200 rpm, in a Clariostar plate reader
952 (BMG Labtech) with OD₆₀₀ read every 10 min for 20 h. To prevent condensation on microplate
953 lids and subsequent interference with readings, lids were treated with 0.05% Triton X-100 in
954 100% ethanol as described previously¹¹⁵. Growth plates had 3 technical replicates, with 3
955 biological replicates for each condition. Area under the curve (AUC) was calculated using the
956 Growthcurver package (v0.3.1) in RStudio with default settings for whole plate growth
957 curves¹¹³. Data processing and graphs were produced in RStudio using the tidyverse package
958 (v1.3.1).

3) The experiments examining metabolic profiles are interesting. If possible, the experiments showing “cross-feeding” would be enhanced by using defined metabolic pathway mutants that cannot utilize certain carbon sources.

- We appreciate the reviewer’s suggestion to test defined metabolic pathway mutants for certain carbon sources. Given the very similar cross-feeding phenotypes of AB6870155 with AB5075 *A. baumannii* strain (see comment 2 above) and our access to a mutant library of AB5075, we have performed successional co-feeding experiments with alcohol dehydrogenase (AB_0191 and AB_3443) and aldehyde dehydrogenase (AB_2971 and AB_1150) mutants of *A. baumannii* AB5075 strain and wild-type AB5075 co-fed with *K. pneumoniae* KP6870155 supernatants after growth in glycerol, lactose, mannitol and sucrose (see below).

- AB_0191 is an ortholog of NH10_00199 which had a 3.2 fold increase in expression in AB6870155 co-cultured with KP6870155, whereas AB_3443 is a second alcohol dehydrogenase in AB5075 and whose ortholog in AB6870155 (NH10_03385) had a 3.5 fold increase in expression. The two alcohol dehydrogenases tested AB_0191 and AB_3443 share a low protein ID (20.7%) and hence their roles may not be redundant.
- Two aldehyde dehydrogenase mutants (AB_2971 and AB_1150) which encode proteins that share a 34.3% protein ID and are both orthologs to AB6870155 aldehyde dehydrogenase (NH10_01161) were also tested. Since AB_2971 and AB_1150 have a higher protein identity, it is possible they have redundant roles and could explain why no impact was observed in either of these single mutants.
- Although a slight decrease in growth of the AB_UW_0191 mutant was observed when co-fed spent media from KP6870155 grown with glycerol, the results from this experiment are not conclusive given the potential complexity of redundancy for these genes within the genome. As such we have found that the result is not conclusive and have therefore not included it in the paper.

4) In Figure 5f there is maybe a 2-fold increase in average biofilm size with co-culture. Is it certain that this difference is biologically relevant?

- We thank the reviewer for this question. Previous work has demonstrated increased average cell length (~2-6 fold) upon exposure to antibiotic stress

(references 48-51). The ~2 fold increase observed (Fig. 5f) corresponds with the significant increase in stress response gene expression revealed by RNA-Seq. We have provided clarification of this biological relevance to the text as follows:

446 lengths (Fig. 5f). Although filamentation of *A. baumannii* and *K. pneumoniae* has been
447 previously observed in cells exposed to various classes of antibiotics ⁴⁸⁻⁵¹, this phenomenon in
448 response to mixed-species co-culturing has not yet been reported for bacteria ⁴². The previously
449 observed filamentation of these species in response to antibiotic stress corresponds with the
450 increased expression of stress response genes observed for both species in the co-culture grown
451 biofilms (Fig. 5a).

5) In Figure 6b the combination of bacteria is more lethal, but twice as many cfu are being inoculated. As a control it is important to test inoculation of 2×10^5 of each single bacterium, to separate the inoculum amount from the single or combined makeup of the inoculum.

- We thank the reviewer for this comment and have repeated the Galleria experiment to test whether the same total inoculum size for single vs combined species shows a difference in virulence. Our results indicate that there is still a clear trend toward higher virulence for coinfection than single infection, even when the final inoculum size is equivalent in the single strain infection as the coinfection. This additional information has been incorporated into the text of the Results and Methods sections as follows and as a Supplementary Figure.

Figure S16 - Effect of inoculum dose of single infection versus coinfection with *A. baumannii* AB6870155 and *K. pneumoniae* KP6870155. Kaplan-Meier curves of single injected AB6870155 (AB) and KP6870155 (KP) and co-injected (A + K) at a 1:1 ratio in *G. mellonella* (p-value based on log-rank test).

Revised results section:

671 injected as single infections (Fig. 6d). This effect was also observed, albeit to a lesser extent,
 672 when the inoculum size of single infection matched that of coinfection (Fig. S16).

Revised methods section:

1044 ¹²⁴. In a separate batch trial, three to four technical replicates comprising of 5 larvae were
 1045 performed using 1 x 10⁷ cells of either strain. For coinfections a mixture of 5 x 10⁶ of each
 1046 AB6870155 and KP6870155 was made prior to injection. Following injection, the larvae were
 1047 incubated at 37°C and examined each day for up to 10 days post-injection and scored for
 1048 survival.

6) Are Klebsiella exposed to Acinetobacter supernatants more antibiotic resistant? Or is the resistance due to mixed biofilm structure?

- We have performed a Millicell hanging insert experiment with cefotaxime to answer this important question. We found that Klebsiella exposed to Acinetobacter supernatants are more antibiotic resistant. This result has been incorporated into the Results, Methods and Supplementary Figure as follows:
- Revised results section:

510 $\mu\text{g/ml}$ to 8 $\mu\text{g/ml}$ (Fig. 6a). To further elucidate the mode of cross-protection, we performed a
 511 Millicell hanging insert experiment where KP6870155 was physically separated from
 512 AB6870155 but could share the same media and any secreted molecules therein. Growth of
 513 KP6870155 was significantly higher when sharing media with AB6870155 than when grown
 514 alone (Fig. S10b). These results indicate there is cross-protection of KP6870155 by
 515 AB6870155, likely through cephalosporinase activity.

Figure S10 – Cross-protection in physically interacting and separated co-cultures of *A. baumannii* and *K. pneumoniae*. **a.** Survival measured by colony counts (CFU/mL) of mono-cultures, AB6870155 (AB) and KP6870155 (KP), and co-cultures of AB6870155 + KP6870155 (AB+KP) at 70:30 ratio after initial (0 hr) and 21 hr of exposure to 512 $\mu\text{g/mL}$ cefotaxime. Red colony counts corresponding to KP6870155 on MacConkey agar in the co-culture plated cells are represented by KP (AB+KP) and white colony counts corresponding to AB6870155 in the co-culture plated cells are represented by AB (AB+KP). **b.** Growth in various concentrations of cefotaxime measured by optical density (OD 600nm) of mono-cultures of KP6870155 (KP only) and physically separated (via Millicell hanging insert) co-cultures of KP6870155 and AB6870155 (KP + AB insert).

- Revised methods section:

794 Cross-protection experiments were also performed using Millicell hanging inserts (Millipore)
 795 where *A. baumannii* AB6870155 was inoculated into inserts that were placed into wells of a
 796 24-well plate inoculated with *K. pneumoniae* KP6870155 at 5 x 10⁵ CFU/ml. Various
 797 concentrations of cefotaxime (16-512 µg/ml) were tested and after 21 hr incubation at 37°C,
 798 cell growth was measured at OD₆₀₀ with a spectrophotometer.

7) The mixed culture resistance was to the level of resistance of the Acinetobacter. Is this a feature of the metabolic interaction of the strains, or just the fact that this one Acinetobacter is more resistant? If a susceptible Acinetobacter strain is used, would there be any protection?

- We thank the reviewer for this question and have expanded the MIC testing to include more susceptible Acinetobacter strains. We have performed MIC testing of the Klebsiella KP6870155 strain in co-culture with *A. baumannii* strains ATCC 17978 and E-072658, which are more susceptible to cefotaxime and found that there was no evidence to support cross-protection when *A. baumannii* is not resistant to cefotaxime. The results of these tests were added to Supplementary Table 12 and the findings were incorporated in the Discussion (see below).

Table S12 - Minimum inhibitory concentrations for mono-cultures and co-cultures of *A. baumannii* (ATCC 17978 and E-072658) and *K. pneumoniae* KP6870155 (µg/mL)

Culture	Cefotaxime MIC (µg/ml)
ATCC 17978	4-8
E-072658	2
KP6870155	16
ATCC 17978 + KP6870155	8
E-072658 + KP6870155	4-8

- Revised discussion section:

728 antibiotic meropenem against *S. aureus*. In this work, we observed that *A. baumannii*
 729 AB6870155 was able to cross-protect *K. pneumoniae* KP6870155 against inhibitory
 730 concentrations of cefotaxime that would normally kill this *K. pneumoniae* strain. However, in
 731 the case where *A. baumannii* is more susceptible to cefotaxime, the MIC of KP6870155 and *A.*

733 *baumannii* cocultures is lower than that of KP6870155 pure cultures (Table S12), indicating
 734 there is a fine balance between which strain provides cross-protection and how much each
 735 strain is benefitting from the interaction. These previous studies and the work here demonstrate
 736 a propensity for *A. baumannii* to shelter rather than kill certain co-habiting pathogens.

8) Figure 6a – What is the proportion of starting Klebsiella cells that survived? The picture is nice but it does not give any indication of the magnitude of the killing experienced by Klebsiella. Thus, we cannot have any measure of the protection conferred upon Klebsiella by Acinetobacter. What happens at later timepoints? Are all the Klebsiella killed? Whether the Klebsiella are truly protected is unclear. Again Figure 6a would need a time zero as well to judge this.

- We have repeated the experiment and added colony count information at timepoint zero and 21 hr. This has been added to Supplementary Figure S10a. We have incorporated this experiment into the Methods and Results sections (see below). Note: Figure S10b was added in response to reviewer 1's comment.

Figure S10 – Cross-protection in physically interacting and separated co-cultures of *A. baumannii* and *K. pneumoniae*. **a.** Survival measured by colony counts (CFU/mL) of mono-cultures, AB6870155 (AB) and KP6870155 (KP), and co-cultures of AB6870155 + KP6870155 (AB+KP) at 70:30 ratio after initial (0 hr) and 21 hr of exposure. Red colony counts corresponding to KP6870155 on MacConkey agar in the co-culture plated cells are represented by KP (AB+KP) and white colony counts corresponding to AB6870155 in the co-culture plated cells are represented by AB (AB+KP). **b.** Growth in various concentrations of cefotaxime measured by optical density (OD 600nm) of mono-cultures of KP6870155 (KP only) and physically separated (via Millicell hanging insert) co-cultures of KP6870155 and AB6870155 (KP + AB insert).

- Revised results section:

501 higher abundance of *K. pneumoniae*, had a surviving population of KP6870155 (Fig. 6a). To
 502 further test for cross-protection in cefotaxime, we exposed mono- and co-cultures of
 503 AB6870155 and KP6870155 to 512 µg/ml cefotaxime and plated the cultures onto MacConkey
 504 agar at time zero and 21 hr post exposure to see whether a heteroresistant population could be
 505 revived. However, no colonies grew on the MacConkey agar plates from *K. pneumoniae* mono-
 506 cultures treated with 512 µg/mL cefotaxime after 21 hr of exposure (Fig. S10a), and hence the
 507 surviving KP6870155 population within the co-cultures were likely not heteroresistant cells.

- Revised methods section:

787 To examine *A. baumannii* AB6870155 cross-protection of *K. pneumoniae* KP6870155 when
 788 the two species have physical contact, pure and co-cultures were exposed to 512 µg/ml
 789 cefotaxime and their survival was measured via colony counts of spread plates at time 0 (upon
 790 initial exposure to cefotaxime) and time 21 hr following 21 hr of static incubation at 37°C.

9) The methods state that for co-culture experiments the different bacteria were mixed at “defined ratios”. In Table 2 this ratio is not mentioned. In the legend for Figure 6a this ratio is not mentioned. It is very difficult to completely understand what is being done in some of these experiments.

- We appreciate the reviewer’s feedback and have added the ratio of coculture to Table 2 and specified the ratio of coculture used in the caption for Figure 6a (see below):

465 **Table 2: Minimum inhibitory concentrations for mono-cultures and co-cultures (1:1) of**
 466 **AB6870155 and KP6870155 (µg/mL).**

Culture	Cefotaxime	Meropenem	Doxycycline	Gentamicin	Ampicillin
AB6870155	1024	0.25-0.5	8-16	512	> 512
KP6870155	16	< 0.125	2-4	< 1	> 512
A+K*	1024	0.5	8	> 512	> 512

468 *A+K = AB6870155 + KP6870155

549 **Fig. 6 – Effects of co-cultures on antibiotic resistance and virulence. a,** Antibiotic cross-
 550 protection assay in 512 µg/mL cefotaxime. A + K denotes AB6870155 and KP6870155 co-
 551 cultures at a 30:70 ratio respectively. Inset of MacConkey agar plate shows KP (KP6870155)

Minor comments

10) The way these sentences are written it sounds like *Klebsiella* is not CRE, but it is. “Dual-infections by *Acinetobacter baumannii* or *Pseudomonas aeruginosa* in patients with carbapenem-resistant Enterobacteriaceae (CRE) have been shown to have increased antibiotic resistance levels and mortality rates compared to single infections. Similarly, increased mortality has also been observed in critically ill patients co-infected with *Klebsiella pneumoniae*, *P. aeruginosa* and/or *A. baumannii*.

- We have revised to make it clearer that *Klebsiella* is indeed CRE:

43 virulence and antimicrobial resistance ^{3,4}. Dual-infections by *Acinetobacter baumannii* or
44 *Pseudomonas aeruginosa* in patients with carbapenem-resistant Enterobacteriaceae (CRE)
45 have been shown to have increased antibiotic resistance levels and mortality rates compared to
46 single infections ⁵. For instance, increased mortality has been observed in critically ill patients
47 co-infected with *Klebsiella pneumoniae*, *P. aeruginosa* and/or *A. baumannii* ⁵. Furthermore,

11) The legend for Figure 5b does not specify which parameter (growth or biofilm mass) is indicated by the bars or the black points.

- We have added this information to the Figure 5 legend as follows:

427 **Fig. 5 – Biofilm properties of *A. baumannii* AB6870155 and *K. pneumoniae* KP6870155**
428 **co-cultures. a,** RNA-seq expression changes (log₂FC) of AB6870155 and KP6870155 co-
429 cultured biofilms relative to their respective mono-cultured biofilms. **b,** Growth (bars) and
430 biofilm formation (points) (measured as OD₅₅₀ after staining biofilms with crystal violet) of
431 various ratios of AB6870155: KP6870155 grown planktonically; biofilm production detected

Reviewers' comments:

Reviewer #1 (Remarks to the Author):

The authors have addressed my previous comments succinctly and completely. I am very pleased with the final product and commend the authors on an expertly performed study. I have no additional comments.

Reviewer #3 (Remarks to the Author):

The authors have significantly improved the manuscript and addressed the majority of the concerns of the reviewers. However, there are still some points that require clarification.

1) In the new Millicell experiment, the results do not directly implicate a secreted cephalosporinase, however that is one possible explanation. In order to clarify that the secreted activity is indeed a cephalosporinase, wouldn't at minimum a beta-lactamase inhibitor like clavulanic acid need to be used to inhibit the activity? This would seemingly provide some specificity.

2) I do not completely understand the co-culture experiment and the explanation the authors provide. If there are aggregates when plating 10 ul drops of culture that make colony counting difficult, the authors can plate each sample/dilution on a separate agar plate - usually 100 ul per plate. Then in theory there should not be problems with aggregation of the colonies on the plate. Or are the authors saying that there are large, noticeable aggregates of cells in liquid culture prior to plating?

3) I do not understand the response to Reviewer 3, point 4. I was previously asking if the modest change in biofilm size was biologically relevant. The authors' answer is that *Acinetobacter* cells can filament and increase in size after treatment with some antibiotics. I do not understand how that addresses the question. Are the authors saying that the increase in biofilm size is due to an increase in the size of each cell rather than an increase in cell number?

4) It should be noted that overall, the difference in virulence phenotype in *Galleria* is very modest. The increase in virulence appears to be consistent with an additive effect. It is not clear that synergy between the two pathogens is driving the modest increase in virulence.

5) It is understandable but unfortunate that the experiments with metabolic pathway transposon mutants were not conclusive. This could have provided a detailed metabolic link to prove that the cross-feeding is really driving a synergistic effect.

Responses to reviewer 3:

1) In the new Millicell experiment, the results do not directly implicate a secreted cephalosporinase, however that is one possible explanation. In order to clarify that the secreted activity is indeed a cephalosporinase, wouldn't at minimum a beta-lactamase inhibitor like clavulanic acid need to be used to inhibit the activity? This would seemingly provide some specificity.

- We appreciate the reviewer's suggestion and performed the Millicell experiment with the beta-lactamase sulbactam instead of clavulanic acid as previous reports have shown synergy between cefotaxime and sulbactam and antagonism with cefotaxime and clavulanic acid. The result directly implicates cephalosporinases are the cross-protecting agent since while *A. baumannii* improves *K. pneumoniae* resistance to cefotaxime, it does not do so for cefotaxime in combination with beta-lactamase inhibitor sulbactam. We additionally treated nitrocefin discs with washed cell pellet vs. filtered supernatant of *A. baumannii* grown with CEF vs CEF:SUL and observed cephalosporinases are found in filtered supernatant for CEF only but not when treated with CEF:SUL. This result has been incorporated into the manuscript as follows:

Revised Figure 6 to incorporate millicell experiment with cefotaxime:sulbactam:

519 **Fig. 6 – Effects of co-cultures on antibiotic resistance and virulence.** a, Antibiotic cross-
 520 protection assay in 512 µg/mL cefotaxime. A + K denotes AB6870155 and KP6870155 co-
 521 cultures at a 30:70 ratio respectively. Inset of MacConkey agar plate shows KP (KP6870155)
 522 colonies appearing red due to *K. pneumoniae* β-galactosidase activity; AB (AB6870155) lacks

524 this enzyme and appears as white colonies. b, Growth of *K. pneumoniae* KP6870155 in
 525 cefotaxime only and cefotaxime + sulbactam CEF:SUL (2:1) as measured by optical density
 526 (OD₆₀₀). “KP only” represents *K. pneumoniae* KP6870155 grown in wells without sharing
 527 media with *A. baumannii* AB6870155 while “KP (+AB insert)” represents KP6870155
 528 physically separated from AB6870155 via a Millicell hanging insert and able to share media.
 529 Significance was measured by two-tailed t-test where * represents a p-value < 0.05 and ns (not
 530 significant) represents a p-value > 0.05. c, Kaplan-Meier curves of single injected AB6870155
 531 (AB) and KP6870155 (KP) and co-injected (A + K) at a 1:1 ratio in *G. mellonella* (p-value
 532 based on log-rank test); PBS only controls had 100% survival (data not shown due to overlap
 533 with AB curve). d, schematic representation of *A. baumannii* AB6870155 and *K. pneumoniae*
 534 KP6870155 interactions.

Revised Figure S10 to incorporate nitrocefin disc test:

Figure S10 – Cross-protection in physically interacting and separated co-cultures of *A. baumannii* and *K. pneumoniae* and secretion of cephalosporinases. a, Survival measured by colony counts (CFU/mL) of mono-cultures, AB6870155 (AB) and KP6870155 (KP), and co-cultures of AB6870155 + KP6870155 (AB+KP) at 70:30 ratio after initial (0 hr) and 21 hr of exposure to 512 µg/mL cefotaxime. Red colony counts corresponding to KP6870155 on MacConkey agar in the co-culture plated cells are represented by KP (AB+KP) and white colony counts corresponding to AB6870155 in the co-culture plated cells are represented by AB (AB+KP). b, nitrocefin discs treated with filter sterilised cell supernatant (cell filtrate) and washed cells (cell pellet) of *A. baumannii* grown in the presence of 32 µg/ml cefotaxime or 16 µg/ml cefotaxime:sulbactam (2:1).

Revised results:

480 AB6870155 but could share the same media and any secreted molecules therein. Growth of
481 KP6870155 was significantly higher when sharing media with AB6870155 than when grown
482 alone (Fig. 6b) in the presence of cefotaxime (CEF) treatment but not when using the beta-
483 lactamase inhibitor, sulbactam, in combination with cefotaxime (CEF:SUL). Cephalosporinase
484 secretion by *A. baumannii* was diminished in the CEF:SUL combined treatment (Fig. S10b)
485 indicating cross-protection of KP6870155 by AB6870155 is via secreted cephalosporinases.

Also revised results section describing figure 6, since the figure was modified to incorporate the new millicell experiment results:

623 most virulent with 7% of larvae dead by day 10, whereas *A. baumannii* AB6870155 was least
624 virulent, with a 100% survival rate (Fig. 6c). Hence the combination of the two strains at the
625 injection dose tested had a synergistic killing effect where the total larvae dead significantly
626 exceeded that of both strains injected as single infections. This effect was also observed, albeit
627 to a lesser extent, when the inoculum size of single infection matched that of coinfection (Fig.

Lucie Semec
Deleted: If the effect of co-infection were simply additive, then the rate of killing would be the sum of total larvae dead per single injection (dashed bar plot in Fig. 6d). However, t

Lucie Semec
Deleted: (Fig. 6d)

Revised methods:

752 24-well plate inoculated with *K. pneumoniae* KP6870155 at 5×10^5 CFU/ml. Various
753 concentrations (8-32 $\mu\text{g/ml}$) of cefotaxime and cefotaxime:sulbactam (2:1 ratios) were tested.
754 Cell growth was measured at OD₆₀₀ with a spectrophotometer after 21 hr incubation at 37°C.
755 To test whether *A. baumannii* cephalosporinases were secreted, cell pellet versus extracellular
756 secreted fractions were prepared and used to treat nitrocefin discs, where appearance of a red
757 colour indicates presence of cephalosporinases. *A. baumannii* grown in the presence of
758 cefotaxime and cefotaxime:sulbactam, normalised to the same OD₆₀₀ was pelleted at 3,000 g
759 for 10 minutes. Supernatants were filter sterilised with a 0.2 μm filter (cell filtrate fraction) and
760 cell pellets were washed twice in 1x PBS with centrifugation of 3,000 g for 10 minutes between
761 washes (cell pellet fraction). A 10 μl drop of cell filtrate and washed cell pellet was placed onto
762 nitrocefin discs and colour change was recorded after 5 minutes of incubation at room
763 temperature.

2) I do not completely understand the co-culture experiment and the explanation the authors provide. If there are aggregates when plating 10 ul drops of culture that make colony counting difficult, the authors can plate each sample/dilution on a separate agar plate - usually 100 ul per plate. Then in theory there should not be problems with aggregation of the colonies on the plate. Or are the authors saying that there are large, noticeable aggregates of cells in liquid culture prior to plating?

- Yes that is correct, there were noticeable aggregates of cells in liquid culture prior to plating (see attached photo) which is why plating on separate plates does not help.

aggregation of AB – in SLMM

3) I do not understand the response to Reviewer 3, point 4. I was previously asking if the modest change in biofilm size was biologically relevant. The authors' answer is that Acinetobacter cells can filament and increase in size after treatment with some antibiotics. I do not understand how that addresses the question. Are the authors saying that the increase in biofilm size is due to an increase in the size of each cell rather than an increase in cell number?

- We had commented on the increase in cell length as the reviewer had referred to Fig. 5f in their comment which plots the distribution of cell lengths between the different samples and has nothing to do with biofilm size. Perhaps the reviewer had misinterpreted that figure 5f was depicting biofilm size, however that is not the case as it is clearly stated in the figure legend that this is depicting mean cell length.

4) It should be noted that overall, the difference in virulence phenotype in Galleria is very modest. The increase in virulence appears to be consistent with an additive effect. It is not clear that synergy between the two pathogens is driving the modest increase in virulence.

- We do not agree that the virulence is consistent with an additive effect as the trend for increased virulence of co-cultures was statistically significant ($p < 0.05$).

5) It is understandable but unfortunate that the experiments with metabolic pathway transposon mutants were not conclusive. This could have provided a detailed metabolic link to prove that the cross-feeding is really driving a synergistic effect.

- Yes, we agree it is understandable but unfortunate, but this is the reality of biology when working with pathways that have redundancy.

REVIEWERS' COMMENTS

Reviewer #3 (Remarks to the Author):

I thank the authors for their explanations and additional experiments. I think the addition of data directly implicating a cephalosporinase is important and significantly enhances the manuscript.